# The cholesterol transporter NPC1 is essential for epigenetic regulation and maturation of oligodendrocyte lineage cells

Thaddeus J. Kunkel[1,5], Alice Townsend[2,5], Kyle A. Sullivan [3], Jean Merlet[2], Edward H. Schuchman[4], Daniel A. Jacobson [3] ✉ & Andrew P. Lieberman [1] ✉

The intracellular cholesterol transporter NPC1 functions in late endosomes and lysosomes to efflux unesterified cholesterol, and its deficiency causes Niemann–Pick disease Type C, an autosomal recessive lysosomal disorder characterized by progressive neurodegeneration and early death. Here, we use single-nucleus RNA-seq on the forebrain of $Npc1^{-/-}$ mice at P16 to identify cell types and pathways affected early in pathogenesis. Our analysis uncovers significant transcriptional changes in the oligodendrocyte lineage during developmental myelination, accompanied by diminished maturation of myelinating oligodendrocytes. We identify upregulation of genes associated with neurogenesis and synapse formation in $Npc1^{-/-}$ oligodendrocyte lineage cells, reflecting diminished gene silencing by H3K27me3. $Npc1^{-/-}$ oligodendrocyte progenitor cells reproduce impaired maturation in vitro, and this phenotype is rescued by treatment with GSK-J4, a small molecule inhibitor of H3K27 demethylases. Moreover, mobilizing stored cholesterol in $Npc1^{-/-}$ mice by a single administration of 2-hydroxypropyl-β-cyclodextrin at P7 rescues myelination, epigenetic marks, and oligodendrocyte gene expression. Our findings highlight an important role for NPC1 in oligodendrocyte lineage maturation and epigenetic regulation, and identify potential targets for therapeutic intervention.

Oligodendrocytes, the myelin-producing cells of the CNS, arise from oligodendrocyte progenitor cells (OPCs), a heterogeneous group of cells that retain some pluripotency and may have varying functions, including antigen presentation[1–7]. As they commit to becoming oligodendrocytes, OPCs differentiate into post-mitotic immature, or pre-myelinating, oligodendrocytes. Immature oligodendrocytes contact axons and further differentiate to become mature, myelinating oligodendrocytes. Oligodendrocyte differentiation and maturation are tightly regulated by both intrinsic and extrinsic signals[8–11].

Cholesterol is rate-limiting in the formation of myelin[12,13]. Myelin is more cholesterol-rich than other plasma membranes, with cholesterol being its most abundant lipid[14–16]. Approximately 80% of the brain's cholesterol is contained within myelin[14,17]. As cholesterol-rich lipoprotein particles do not cross the blood-brain barrier, most cholesterol used in myelin formation is synthesized de novo. Cholesterol synthesized by both oligodendrocytes and neighboring cells is necessary for myelination[14,15,18,19]. Notably, the mechanism by which cholesterol promotes myelination and its potential influence on oligodendrocyte differentiation remains poorly understood.

[1]Department of Pathology, University of Michigan Medical School, Ann Arbor, MI, USA. [2]The Bredesen Center for Interdisciplinary Research and Graduate Education, University of Tennessee Knoxville, Knoxville, TN, USA. [3]Computational and Predictive Biology, Oak Ridge National Laboratory, Oak Ridge, TN, USA. [4]Department of Genetics and Genomic Sciences, The Icahn School of Medicine at Mount Sinai, New York, NY, USA. [5]These authors contributed equally: Thaddeus J. Kunkel, Alice Townsend. ✉e-mail: jacobsonda@ornl.gov; liebermn@umich.edu

Cholesterol-rich lipoprotein particles enter cells through receptor-mediated endocytosis and transit to late endosomes and lysosomes. There, the NPC1 and NPC2 proteins work together to efflux unesterified cholesterol for use in cellular processes such as sterol biosynthesis and membrane integration[20–23]. Mutations in these proteins lead to Niemann–Pick disease Type C, a rare autosomal recessive lysosomal disorder characterized by the accumulation of unesterified cholesterol in late endosomes and lysosomes[24–26]. Neurons of the CNS are particularly vulnerable to this lipid trafficking defect, as progressive neurodegeneration is a cardinal feature of Niemann–Pick Type C. Accumulating evidence indicates that oligodendrocyte dysfunction also contributes to the neurological phenotype that occurs in this disorder. Individuals with Niemann–Pick Type C have myelin deficits, most prominently in the major white matter tracts of the brain, where the extent of myelin depletion positively correlates with clinical severity[27–29]. This is recapitulated in mice homozygous for a null allele of the $Npc1$ gene ($Npc1^{-/-}$), where developmental hypomyelination is observed prior to neuron loss[30–32]. This developmental hypomyelination is seen throughout the brain but is most severe in the mouse forebrain[30,32]. Furthermore, deletion of $Npc1$ only in oligodendrocytes leads to myelin deficits, neurodegeneration, and ataxia, demonstrating a role for oligodendrocytes in Niemann–Pick Type C pathogenesis[30]. Interestingly, deletion of $Npc1$ specifically in neurons also leads to robust myelination defects, illustrating that both cell-autonomous and non-cell-autonomous pathways are involved in the hypomyelination phenotype[30].

In this study, we used single-nucleus RNA sequencing and chromatin immunoprecipitation (ChIP) in combination with targeted assays to analyze the pathways underlying developmental hypomyelination in $Npc1^{-/-}$ mice. Because the forebrain is where hypomyelination is most severe and single-cell sequencing was performed previously on $Npc1^{-/-}$ mouse cerebella[33], we focused our studies on the forebrain. We show that in the early postnatal period, cells in the oligodendrocyte lineage are the most affected cell types in the forebrain, undergoing striking transcriptional changes and cell death. Furthermore, we demonstrate that epigenetic dysfunction is a key contributor to oligodendrocyte dysfunction in $Npc1^{-/-}$ mice. Lastly, we show that these deficits in oligodendrocyte differentiation and survival are tied to the bioavailability of cholesterol. This work furthers our understanding of the role of cholesterol in regulating oligodendrocyte differentiation and provides targets for potential therapeutic intervention for the treatment of Niemann–Pick Type C.

## Results

### The oligodendrocyte lineage is severely affected in $Npc1^{-/-}$ mice during development

To identify the cell types and pathways in the brain that are most affected in early Niemann–Pick C pathogenesis, we performed single-nucleus RNA-sequencing (snRNA-seq) on the forebrain of littermate WT and $Npc1^{-/-}$ mice at 16 days of age (P16) using three biological replicates for each genotype. This unbiased analysis complemented our previous work, which demonstrates phenotypic changes in glia at this age, even in the absence of neuron death[30]. Approximately 13,000 nuclei from each mouse were sequenced at a depth of ~80,000 reads per cell. Expression patterns of previously established marker genes were used to identify cell type-specific clusters[34–36] (Fig. 1A, B and Supplementary Fig. 1). Gene expression analysis revealed thousands of differentially expressed genes between WT and $Npc1^{-/-}$ samples (Fig. 1C). Notably, immature oligodendrocytes had 1764 differentially expressed genes, the most of any cluster (Fig. 1C and Supplementary Data 1). Our analysis separated the oligodendrocyte lineage into three main stages: proliferative oligodendrocyte progenitor cells (OPCs), post-mitotic and actively differentiating immature oligodendrocytes, and myelin-forming mature oligodendrocytes. Differential cell abundance analysis found significant decreases in immature and mature

oligodendrocytes in $Npc1^{-/-}$ samples compared to WT, while all other cell populations were present in normal proportions (Fig. 1D and Supplementary Table 2). This is in accordance with published data showing severe hypomyelination and a lack of mature, myelin-forming oligodendrocytes in $Npc1^{-/-}$ mice during development[30,32,37]. This hypomyelination can be readily visualized by immunohistochemical staining for myelin basic protein (MBP) in the corpus callosum of P16 mice (Fig. 2A), corroborating previously published findings[30,32,37]. Taken together, these data indicate that the oligodendrocyte lineage is markedly impacted at an early timepoint in the Niemann–Pick C brain and highlight a particular vulnerability in immature oligodendrocytes.

To further explore the reduction of oligodendrocyte lineage cells in $Npc1^{-/-}$ mice, we stained for OLIG2, a transcription factor found throughout the lineage[35,36]. The number of OLIG2-positive cells in the corpus callosum was quantified at three ages: P6, P9, and P16. We chose these ages as they roughly correspond to sequential stages of oligodendrocyte development[17,38–40]. At P6, OPCs are present in large numbers and remain largely non-differentiated. At P9, active differentiation has begun and there is an increase in immature oligodendrocytes. By P16, there is peak myelination with large numbers of mature cells[17,38–40]. OLIG2-positive cell numbers were comparable between $Npc1^{-/-}$ and WT mice at P6, suggesting similar numbers of OPCs in these mice. However, by P9, $Npc1^{-/-}$ mice had significantly reduced numbers of OLIG2-positive cells compared to WT, and by P16, this reduction was even more pronounced (Fig. 2B). These data indicate a reduction in the number of oligodendrocyte lineage cells as they differentiate into immature and mature oligodendrocytes, a finding that is in line with the quantification of cell numbers from snRNA-seq (Fig. 1D). Correspondingly, western blot of forebrain lysates showed reductions in the oligodendrocyte lineage transcription factors OLIG2 and SOX10[35,36] in $Npc1^{-/-}$ mice at P16, but not at P6 (Fig. 2C). We conclude that there is a gradual reduction in oligodendrocyte lineage markers in the Niemann–Pick C brain as these cells undergo differentiation. In contrast, neurofilament (NF200) levels (Fig. 2C) and neuron numbers (Fig. 1D and Supplementary Table 1) were similar between WT and $Npc1^{-/-}$ mice, indicating that deficits in the oligodendrocyte lineage were not secondary to neuron loss. Notably, there was no significant difference in the proliferation of oligodendrocyte lineage cells in the corpus callosum of mice at P6, P9, or P16 as assessed by staining for Ki67[41] (Supplementary Fig. 2). However, terminal deoxynucleotidyl transferase dUTP nick end labeling (TUNEL) staining[42] revealed a significant increase in cell death in the corpus callosum of $Npc1^{-/-}$ mice at P9 when compared to WT. This increase in cell death was not apparent at P6 or P16 (Fig. 2D). Furthermore, we performed qRT-PCR to measure the expression of marker genes specific to OPCs ($Cspg4$ and $Smoc1$), immature oligodendrocytes ($Enpp6$ and $Dusp15$), and mature oligodendrocytes ($Mbp$ and $Plp1$)[34–36,43] in the forebrain of an independent cohort of mice at P16. OPC markers were expressed at similar levels in WT and $Npc1^{-/-}$ forebrain (Fig. 2E). However, immature and mature oligodendrocyte markers were significantly reduced in $Npc1^{-/-}$ mice (Fig. 2E). Taken together, these data strongly suggest that the decrease in oligodendrocyte lineage cells in the Niemann–Pick C brain is due, in part, to cell death, particularly while these cells are at an immature stage and actively differentiating. Notably, diminished numbers of OLIG2-positive cells were most robust in the forebrain, as the number of OLIG2-positive cells in the cerebellar white matter, where myelination defects are milder[30,32], were comparable between WT and $Npc1^{-/-}$ mice (Supplementary Fig. 3).

### Neuronal genes are up-regulated in $Npc1^{-/-}$ immature oligodendrocytes

To better understand the pathways associated with oligodendrocyte dysfunction and loss, we performed Gene Ontology (GO) analysis for biological process terms using differentially expressed genes in oligodendrocyte lineage cells from the snRNA-seq data. As expected,

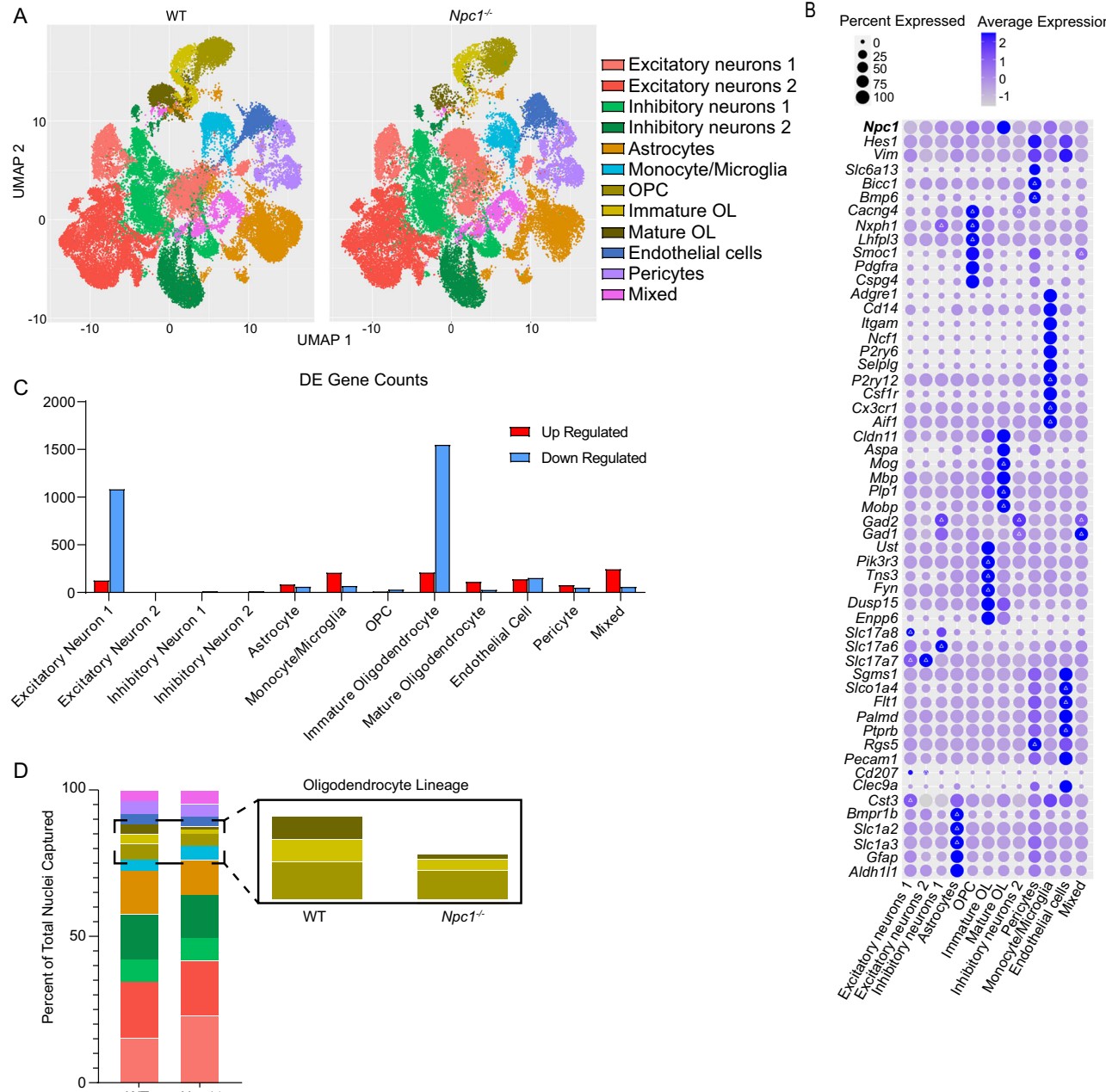

**Fig. 1 | snRNA-seq of WT and *Npc1*$^{-/-}$ forebrain at P16.** Nuclei were isolated from the forebrains of three mice per group and sequenced using the 10X Genomics Chromium platform. **A** UMAP plot for WT and *Npc1*$^{-/-}$ samples with each point representing a single nucleus. Plot points are clustered into cell type based on gene expression patterns. **B** Bubble matrix showing the expression of *Npc1* (at top) and cluster-defining genes. Delta symbols (Δ) within bubbles indicate the three most highly expressed defining genes for each cluster. Size of the bubble corresponds to the percentage of cells expressing a gene, while bubble color indicates expression level. **C** Number of up- (red) and down- (blue) regulated differentially expressed (DE) genes (genes with |log$_2$(fold-change)| of > 0.5 and adjusted $p < 0.05$) per cell cluster as determined through differential expression analysis utilizing a Wilcoxon rank-sum test. Source data are provided as a Source Data file. **D** Graph showing the percentage of the total nuclei captured in each cluster. The colors and cell types correspond with (**A**). Inset shows oligodendrocyte lineage.

genes associated with myelination, axon ensheathment, and oligodendrocyte development were downregulated at each stage of the oligodendrocyte lineage (Fig. 3A, B and Supplementary Data 2). The cholesterol biosynthetic pathway, needed for myelination, was also downregulated in immature oligodendrocytes of mutant mice. Unexpectedly, genes associated with neurogenesis, ion transport, and synapse formation were upregulated in immature and mature oligodendrocytes of Niemann–Pick C mice (Fig. 3A, B and Supplementary Data 2). OPCs also show an upregulation of genes associated with neuron projection morphogenesis (Fig. 3A), suggesting that these pathways are altered early in the lineage. This conclusion is supported by the fact that of the 48 genes that are differentially expressed in OPCs, 42 are significantly altered in immature oligodendrocytes (Supplementary Data 1).

### Repressive histone modifications are decreased in *Npc1*$^{-/-}$ oligodendrocytes

Neuron-associated genes are typically expressed in early OPCs but are silenced during oligodendrocyte differentiation[44–46]. The repressive histone modifications H3K9me3 and H3K27me3 are responsible for epigenetically silencing the expression of neuronal genes in oligodendrocyte lineage cells during normal maturation and

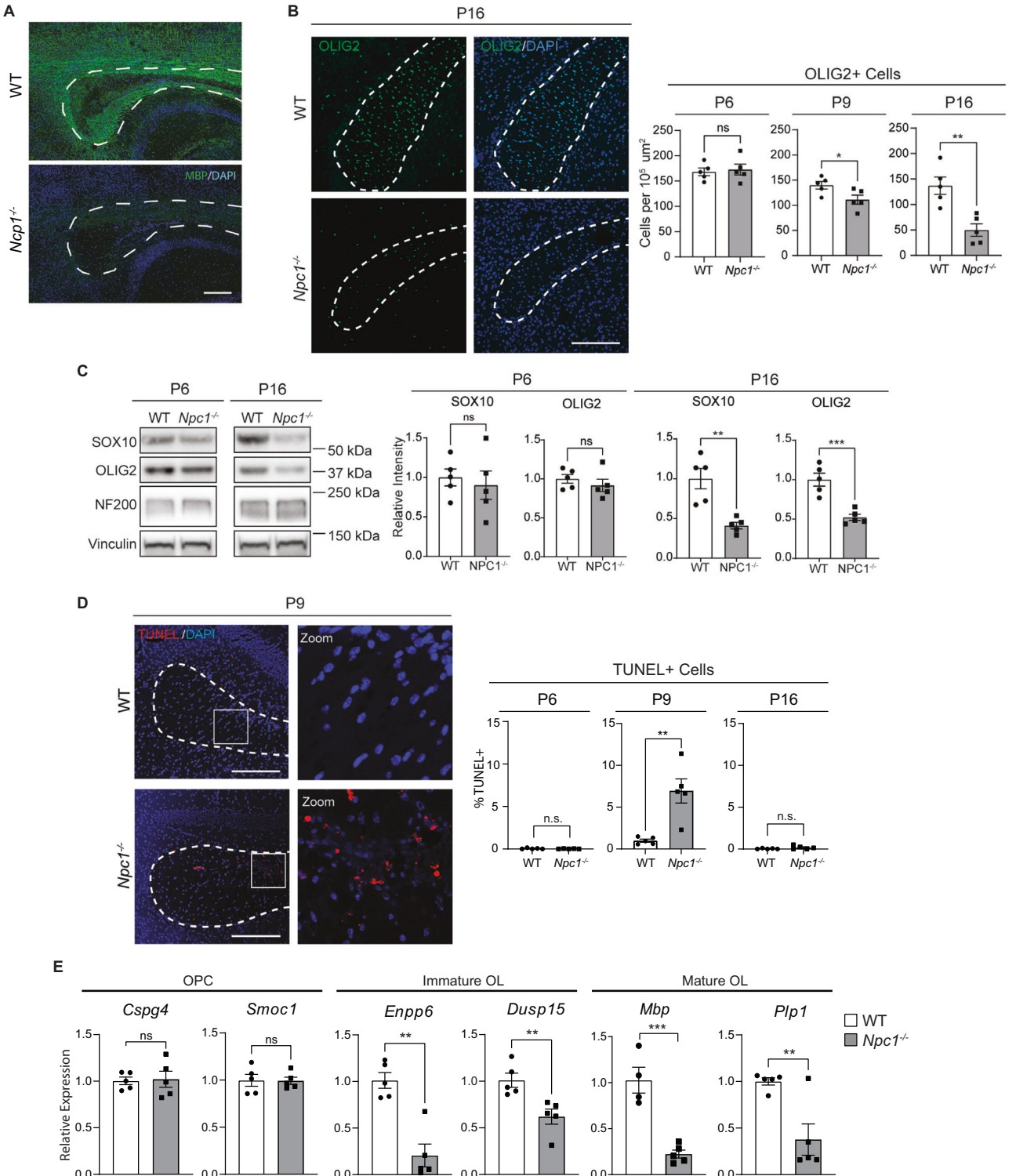

**Fig. 2 | Death of oligodendrocyte lineage cells during development. A** Midline sagittal brain section from P16 mice stained for MBP (green) and DAPI (blue). **B** Brain sections from P6, P9, and P16 WT and *Npc1⁻/⁻* mice were stained for oligodendrocyte lineage cells using OLIG2 (green). Nuclei are stained with DAPI (blue). Quantified at right. **C** Western blot of SOX10, OLIG2, and neurofilament (NF200) from P6 and P16 mouse forebrain. Values were normalized to vinculin and quantified relative to WT. Quantifications at right. **D** Brain sections were stained for DNA double-strand breaks using a TUNEL assay (red) to detect cell death. Nuclei are stained with DAPI (blue). **E** Expression of oligodendrocyte stage-specific markers in P16 mouse forebrain as measured by qRT-PCR and normalized to WT. Dashed lines indicate corpus callosum in all images. Scale bars = 200 μm (**A**) and 150 μm (**B** and **D**). Quantified at right. Data are mean ± SEM. n.s. not significant, *p < 0.05, **p < 0.005, ***p < 0.001 by two-tailed unpaired Student's *t*-test; (**A**) Experiment was done on 3 biological replicates per group, all with similar results; (**B**) *n* = 5 mice; *t* = 0.3385, 2.477, 4.182; df = 8; *p* = 0.7437, 0.0383, 0.0031 (**C**) *n* = 5 mice; *t* = 0.47, 0.90, 4.33, 5.30; df = 8; *p* = 0.6487, 0.3958, 0.0025, 0.0007; (**D**) *n* = 5 mice; *t* = 0.6961, 4.081, 1.463; df = 8; *p* = 0.5061, 0.0035, 0.1817; (**E**) *n* = 5 mice; *t* = 0.1813, 0.06927, 3.891, 3.479, 6.470, 3.618; df = 8; *p* = 0.8606, 0.9465, 0.0046, 0.0083, 0.0002, 0.0068. Source data are provided as a Source Data file.

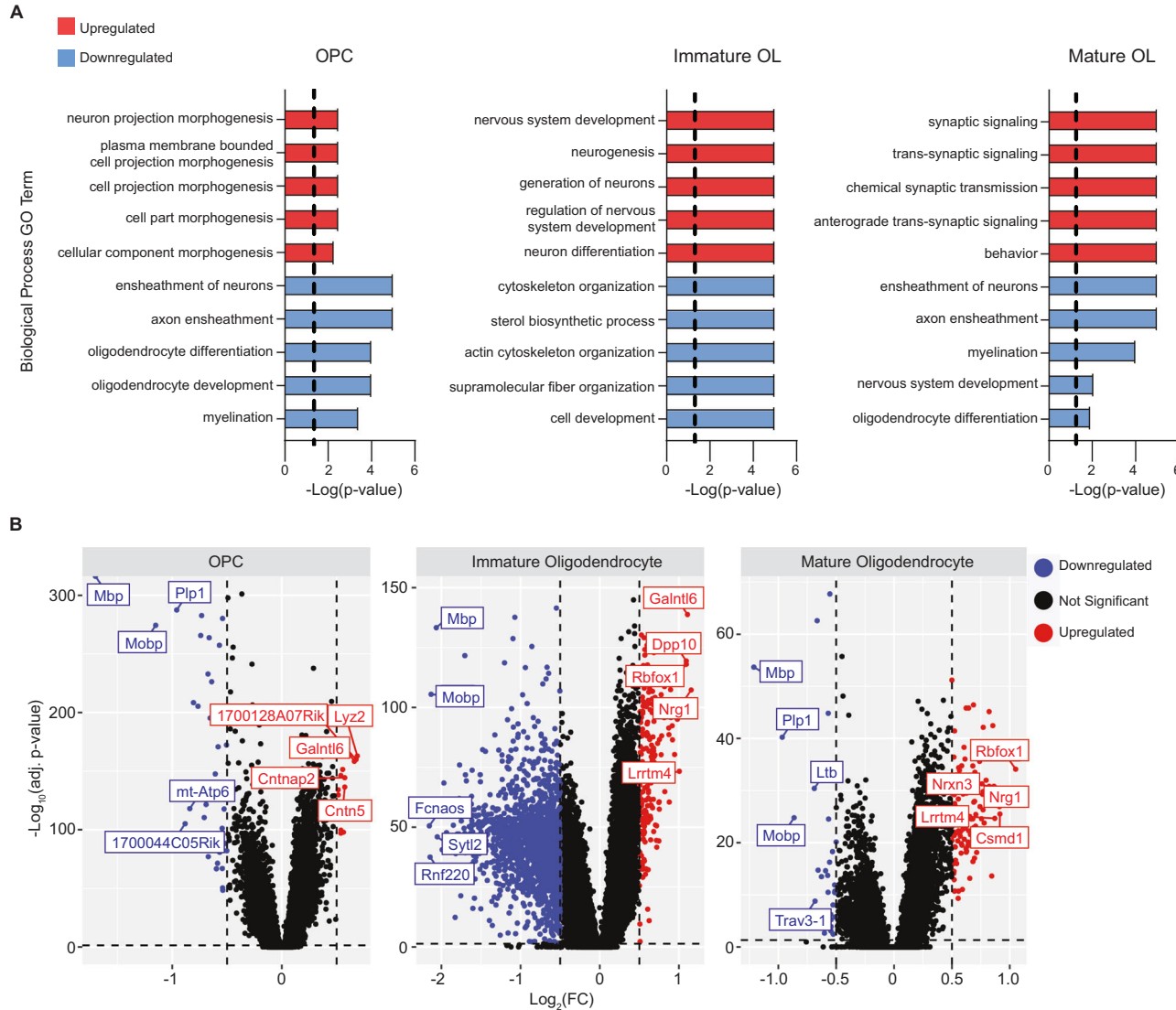

**Fig. 3 | Gene expression changes in the oligodendrocyte lineage. A** Gene Ontology analysis was performed on differentially expressed genes (DE) identified by snRNA-seq. The 10 most significant (by *p*-value) up- (red) and down-regulated (blue) GO terms (biological process) in *Npc1*[−/−] oligodendrocyte lineage cells are shown. The dashed line indicates an adjusted *p* < 0.05 after Bonferroni correction. **B** Volcano plots of DE genes in the oligodendrocyte lineage as determined by differential expression analysis of snRNA-seq. The top 5 up- or down-regulated genes by log$_2$(FC) are labeled. Dashed lines correspond to an adjusted *p*-value of 0.05 and a |log$_2$(FC)| of 0.5.

myelination[44,47]. Furthermore, these histone modifications increase as oligodendrocytes differentiate and are known to play a crucial role in oligodendrocyte differentiation[44,48] (Fig. 4A). Based on gene expression patterns observed in the snRNA-seq dataset, we hypothesized that broad changes in H3K9me3 and H3K27me3 were underlying the transcriptional differences found between oligodendrocyte lineage cells in WT and *Npc1*[−/−] mice. To test our hypothesis, we first stained brain sections for H3K27me3 and H3K9me3 and quantified their intensity. By co-staining for either NeuN or SOX10, we measured these histone modifications in neurons and oligodendrocyte lineage cells, respectively. We identified reduced staining intensity of both H3K27me3 and H3K9me3 in oligodendrocyte lineage cells in the corpus callosum of *Npc1*[−/−] mice compared to WT (Fig. 4B and Supplementary Fig. 4A). Notably, there was no difference in the staining intensity of either histone modification in cortical neurons of WT and *Npc1*[−/−] mice (Fig. 4B and Supplementary Fig. 4B), suggesting the occurrence of oligodendrocyte lineage-specific deficits in these repressive chromatin modifications. These deficits correlated with the severity of developmental dysmyelination, as we did not detect

significant alterations in H3K27me3 in SOX10-positive cells in the cerebellar white matter (Supplementary Fig. 5).

We next assessed histone methylation in the oligodendrocyte lineage by western blot. To acutely isolate OPCs and immature oligodendrocytes from the whole brain of P16 mice, we used magnetic-activated cell sorting (MACS) with microbeads targeting either PDGFRα (expressed in early OPCs) or O4 (expressed later in OPCs and immature oligodendrocytes) (Fig. 4A)[35,49,50]. *Npc1*[−/−] PDGFRα-isolated OPCs showed significantly reduced H3K27me3, but not H3K9me3, compared to WT (Fig. 4C). However, O4-isolated cells showed decreases in both H3K9me3 and H3K27me3 (Fig. 4D), suggesting that the reduction of H3K27me3 may precede H3K9me3 during oligodendrocyte lineage maturation. Western blots of the flow-through cells (cells not bound by microbeads during MACS) showed no changes in H3K27me3 or H3K9me3, further reinforcing that these epigenetic deficits were oligodendrocyte lineage-specific (Fig. 4E).

We next probed our snRNA-seq dataset for potential causes of this epigenetic dysregulation. We found that the expression of *Ezh2*, a

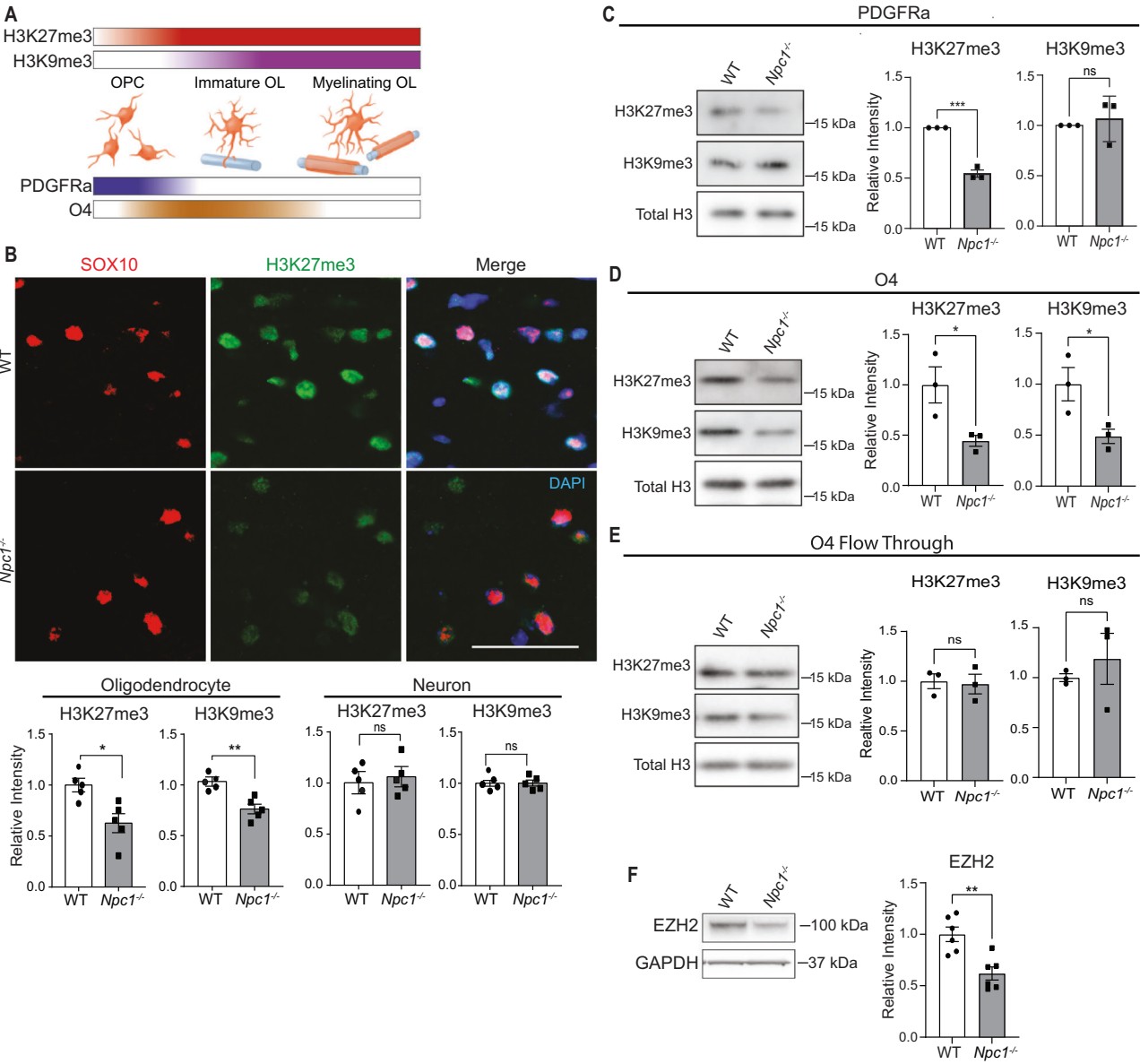

**Fig. 4 | Epigenetic dysregulation of *Npc1*⁻/⁻ oligodendrocytes. A** H3K27me3 and H3K9me3 increase as oligodendrocytes differentiate. PDGFRa expression peaks earlier in development compared to O4. The color in the bars approximates marker levels. **B** Brain sections from P16 WT and *Npc1*⁻/⁻ mice were stained for H3K9me3 or H3K27me3 (green) and co-stained for SOX10 (red, oligodendrocytes) or NeuN (neurons). Nuclei stained with DAPI (blue). Images were captured from the corpus callosum (oligodendrocytes) and cortex (neurons). Representative images show H3K27me3 in OLs; images of NeuN and OL H3K9me3 stains in Supplementary Fig. 4. Scale bar = 50 μm. Methylation staining intensity in the cortex (neurons) and corpus callosum (oligodendrocytes) is quantified below. Methylation intensity was quantified in SOX10+ or NeuN+ cells and values were normalized to WT. Western blot of

histones extracted from cells purified from P16 mice using microbeads targeting PDGFRa (**C**), O4 (**D**), or the flow through cells left over after cell purification (**E**). Values were normalized to total H3 and quantified relative to WT. **F** Western blot of EZH2 from O4-purified cells. Data are mean ± SEM. n.s. not significant, *$p < 0.05$,**$p < 0.005$, ***$p < 0.001$ by two-tailed unpaired Student's *t*-test; (**B**) $n = 5$ mice; $t = 3.1, 4.2, 0.4, 0.3$; df = 8; $p = 0.0146, 0.0031, 0.6765, 0.8028$; (**C**) $n = 3$ biological replicates; $t = 12.9, 0.5$; df = 4; $p = 0.0002, 0.6418$; (**D**) $n = 3$; $t = 3.0, 2.9$; df = 4; $p = 0.0415, 0.0451$ (**E**) $n = 3$ biological replicates; $t = 0.03, 0.7$; df = 4; $p = 0.9754, 0.5059$; (**F**) $n = 6$ biological replicates; $t = 4.0$; df = 10; $p = 0.0025$. Source data are provided as a Source Data file.

histone methyltransferase that catalyzes H3K27 methylation, was reduced in immature oligodendrocytes of Niemann–Pick C mice (Log₂(FC) = −0.726; Supplementary Data 1). Furthermore, western blot analysis of O4-isolated cells revealed reduced levels of EZH2 in *Npc1*⁻/⁻ cells compared to WT (Fig. 4F). This EZH2 reduction and the early appearance of an observable decrease in H3K27me3 led us to focus on H3K27me3 for further investigations.

### Epigenetic dysregulation at neuronal genes in *Npc1*⁻/⁻ cells

ChIP-seq and parallel bulk RNA-seq were performed on O4-purified cells from WT and *Npc1*⁻/⁻ mouse whole brain to identify genes directly

impacted by differential epigenetic regulation. ChIP was conducted for H3K27me3 and H3K27 acetylation (H3K27ac). *Npc1*⁻/⁻ cells had a broad decrease in H3K27me3 across promoter regions (Fig. 5A). Pairing this with RNA-seq data, we identified 66 genes with both a notable decrease in H3K27me3 (Log₂(FC) < −0.3) in their promoter regions and significantly increased gene expression (adjusted $p < 0.05$) (Fig. 5B). GO Term: Cellular Component analysis of these genes revealed three significant terms – "dendrite", "dendritic tree", and "neuron projection" (Fig. 5C). This enrichment was attributable to 16 genes related to neuron projections and synapse formation (Supplementary Table 2), in line with our hypothesis that increased neuron gene expression in

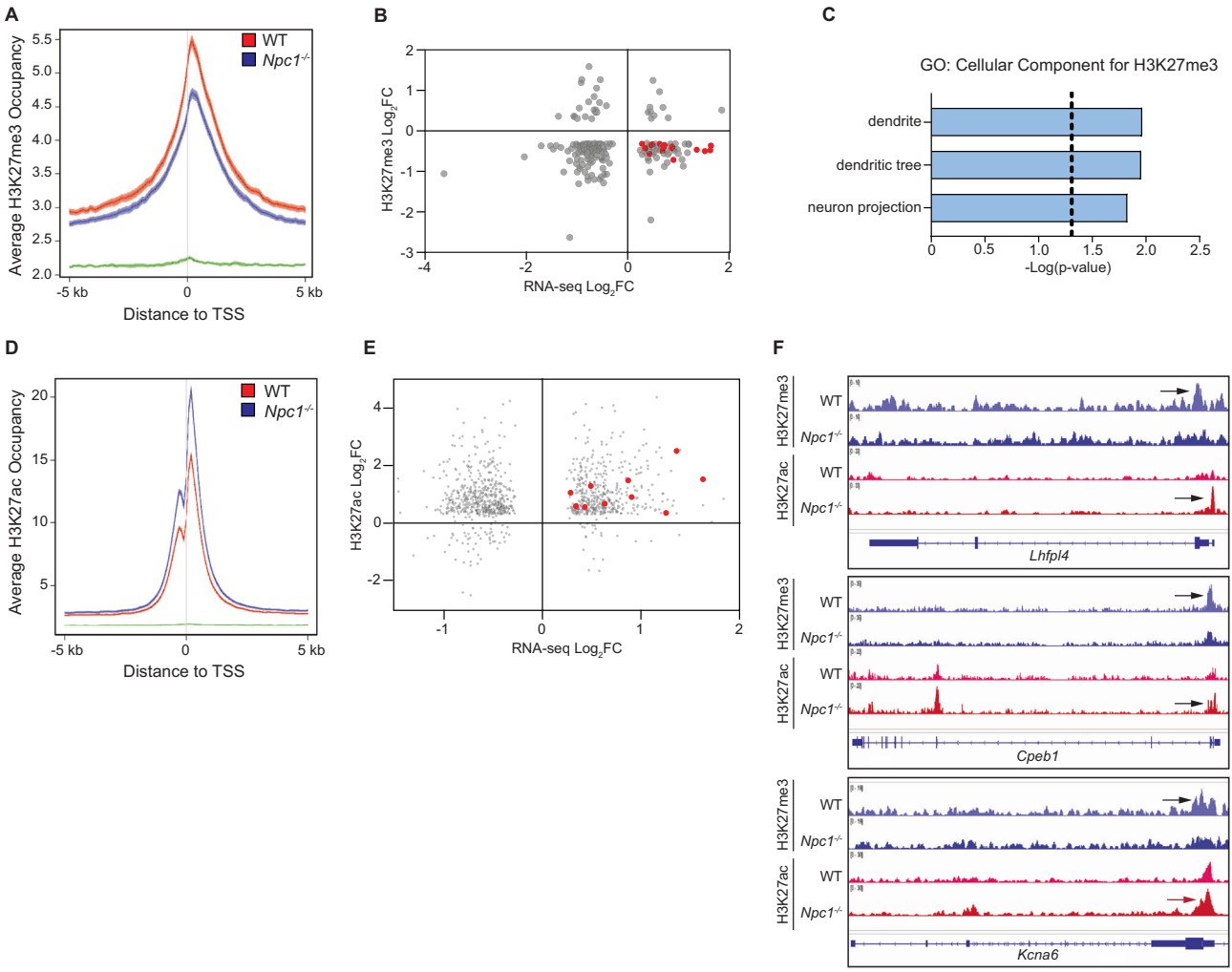

**Fig. 5 | Dysregulation of H3K27me3 and H3K27ac at neuron-related genes.** ChIP-seq was performed on O4-purified cells from P16 WT and *Npc1*[-/-] mice. **A** Average H3K27me3 occupancy at gene promoters (defined as being within 5 kb of transcription start site (TSS)) for WT (red), *Npc1*[-/-] (blue), and background (green). **B** H3K27me3 promoter enrichment (*Npc1*[-/-] relative to WT, log$_2$(fold change), $y$-axis) plotted against gene expression (*Npc1*[-/-] relative to WT, log$_2$(fold change), $x$-axis) for genes with differential H3K27me3 (defined as having a log$_2$(fold change) <−0.3) and significant changes in expression. Red dots indicate 16 genes associated with neuronal pathways. **C** Pathways significantly enriched among genes with decreased H3K27me3 and increased expression in *Npc1*[-/-] cells, as determined by

Gene Ontology: Cellular Component analysis with Bonferroni correction. **D** Average H3K27ac occupancy at gene promoters for WT (red), *Npc1*[-/-] (blue), and background (green). **E** H3K27ac promoter enrichment (*Npc1*[-/-] relative to WT, log$_2$(fold change), $y$-axis) plotted against gene expression (*Npc1*[-/-] relative to WT, log$_2$ fold change, $x$-axis) for genes with differential H3K27ac and significant changes in expression. Red dots indicate 10 genes associated with neuronal pathways. **F** IGV genome browser snapshots of H3K27me3 (blue) and H3K27ac (red) bigWig files at promoter regions of *Lhfpl4*, *Cpeb1*, and *Kcna6*. Arrows point to areas of enrichment. **A**, **D** Center of plotted lines indicates average occupancy, while line thickness shows the 95% confidence interval.

*Npc1*[-/-] oligodendrocyte lineage cells was a consequence of decreased H3K27me3.

Our findings were corroborated by the ChIP-seq-based identification of broad increases in H3K27ac in the promoter regions of genes from *Npc1*[-/-] cells (Fig. 5D). H3K27ac often negatively correlates with H3K27me3, leading to gene activation[51,52]. Correlation with RNA-seq data identified 435 genes with increased promoter region acetylation and significantly higher expression (Fig. 5E). GO Term: Cellular Component gene set enrichment analysis revealed "synapse" as a significantly enriched component in this gene list (Bonferroni-corrected $p = 1.324$E-4). Furthermore, 10 of the 16 up-regulated neuron-related genes with decreased promoter H3K27me3 also exhibited increased promoter H3K27ac (Fig. 5E, F and Supplementary Table 2). Taken together, these data demonstrate that many of the transcriptional changes detected in *Npc1*[-/-] oligodendrocytes, particularly the increases in neuronal gene expression, are the direct result of epigenetic dysregulation.

## The histone demethylase inhibitor GSK-J4 rescues oligodendrocyte differentiation in vitro

We next sought to determine whether epigenetic differences were a cause of impaired oligodendrocyte differentiation or merely a consequence. To answer this question, primary OPCs were purified from whole brains of P6 mice using PDGFRα-targeting microbeads. To promote differentiation, these cells were grown on glass coverslips for 4 days in media containing triiodothyronine. Cultured *Npc1*[-/-] OPCs recapitulated the maturation defect identified in the intact mouse brain (Fig. 6). To target epigenetic marks, we treated OPCs with GSK-J4, a well-characterized selective inhibitor of the H3K27 demethylases KDM6A and KDM6B[53]. Immunocytochemistry revealed that treatment of cultured *Npc1*[-/-] OPCs with GSK-J4 increased H3K27me3 compared to control-treated cells (Fig. 6A). Correspondingly, multiple genes associated with neurogenesis that were upregulated in O4-purified cells due to diminished H3K27me3 (Fig. 5F) had reduced expression after treatment with GSK-J4 (Fig. 6B). Moreover, cells were stained for

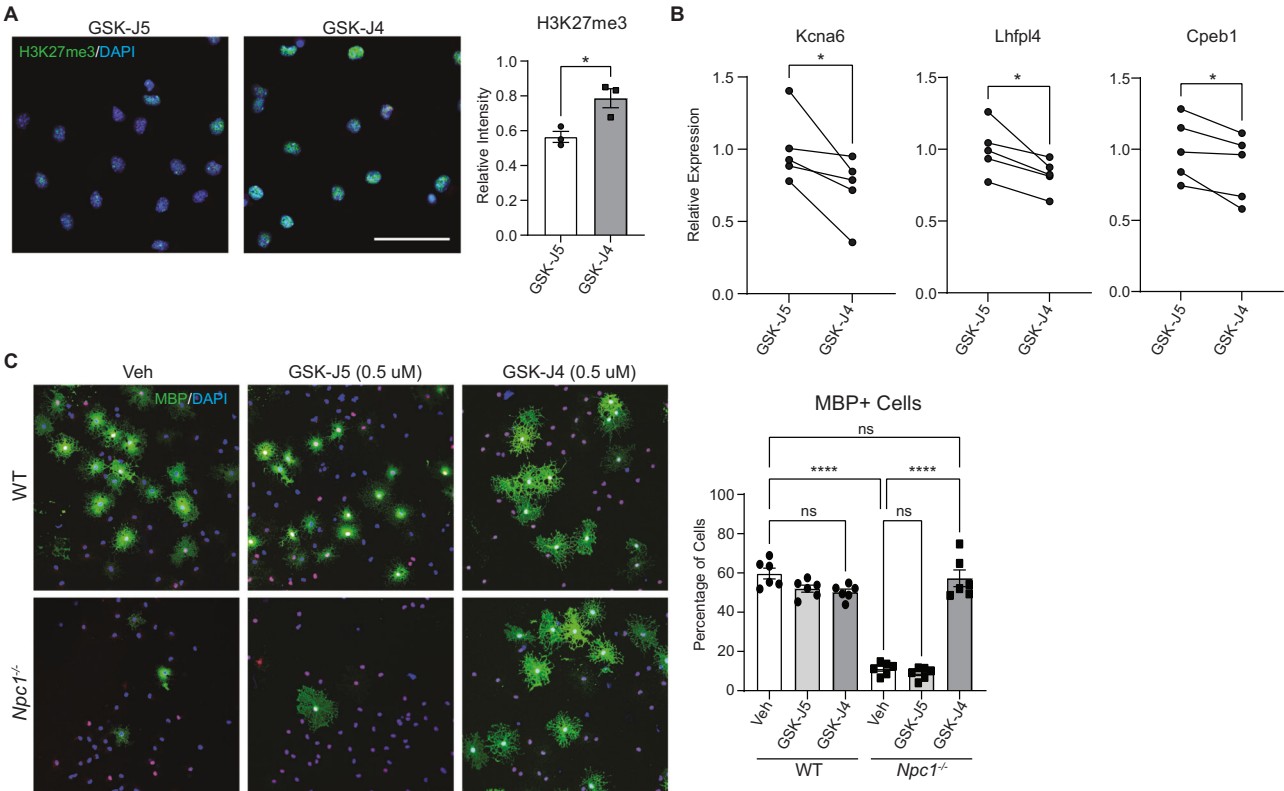

**Fig. 6 | Inhibiting H3K27 demethylase promotes oligodendrocyte differentiation in vitro.** Primary OPCs from WT and *Npc1*[-/-] mice were treated with the KDM6A and KDM6B inhibitor GSK-J4 (0.5 μM), the inactive control GSK-J5 (0.5 μM), or DMSO vehicle (Veh) for 4 days. **A** *Npc1*[-/-] cells treated with GSK-J5 or GSK-J4 and stained for H3K27me3 (green) and DAPI (blue). Intensity of H3K27me3 staining was measured relative to WT cells treated with GSK-J5. **B** *Npc1*[-/-] cells were treated with GSK-J5 or GSK-J4, and gene expression was quantified by qRT-PCR relative to cells treated with GSK-J5. Lines connect paired samples from biological replicate cultures. **C** Cells were stained for SOX10 (red) and MBP (green) to assess differentiation. Nuclei stained with DAPI (blue). The percentage of MBP-positive cells is quantified at the right. Scale bars = 50 μM (**A**) and 100 μM (**C**). Data are mean ± SEM (**A**, **C**). n.s. not significant, *$p < 0.05$, ****$p < 0.0001$ by two-tailed unpaired Student's *t*-test (**A**), two-tailed paired Student's *t*-test (**B**), and one-way ANOVA with Tukey post hoc test and multiple comparisons (**C**). **A** $n = 3$ biological replicates; $t = 3.510$; df = 4; $p = 0.0247$ (**B**) $n = 5$ biological replicates; $t = 2.782, 3.479, 3.158$; df = 4; $p = 0.0497, 0.0254, 0.0343$; (**C**) $n = 6$ biological replicates; $F = 95.44$; df = 5. Source data are provided as a Source Data file.

MBP as an assessment of maturation. Treatment of *Npc1*[-/-] OPCs with GSK-J4 rescued cell maturation to wild type levels (Fig. 6C). In contrast, its inactive isomer, GSK-J5, had no effect (Fig. 6C). Neither small molecule altered maturation of WT OPCs in vitro. These data indicate that altered epigenetic regulation and diminished H3K27 methylation are drivers rather than consequences of impaired oligodendrocyte lineage maturation.

**The Niemann–Pick C myelination phenotype is a consequence of cholesterol mistrafficking**
We sought to determine the role of intracellular lipid trafficking defects in the *Npc1*[-/-] myelination phenotype. Loss of functional NPC1 leads to the lysosomal accumulation of cholesterol plus numerous secondary lipids, including sphingomyelin and complex gangliosides[54–56]. To begin to assess the contribution of these secondary lipid accumulations to the hypomyelination phenotype in *Npc1*[-/-] mice, we evaluated myelination in a mouse model of Niemann–Pick type A disease. These mice are deficient in the gene encoding acid sphingomyelinase (*Smpd1*) and develop progressive neurodegeneration while accumulating sphingomyelin in late endosomes and lysosomes[57,58]. The pathogenic cascade in *Smpd1*[-/-] mice has many similarities with that in *Npc1*[-/-] mice, including the accumulation of secondary lipids, defects in autophagy, lysosomal membrane permeabilization, and cell death[59,60]. Individuals with acid sphingomyelinase deficiency have reduced myelin, and *Smpd1*[-/-] mice show myelin reduction by 4 weeks of age[57,61]. However, in contrast to *Npc1*[-/-] mice,

*Smpd1*[-/-] mice did not display dysmyelination at P16 (Supplementary Fig. 6), suggesting that the developmental myelination defect in *Npc1*[-/-] mice results from disruptions in intracellular cholesterol trafficking rather than from sphingolipid accumulation or lysosomal perturbations.

To directly test if mobilizing intracellular cholesterol improves developmental myelination, we treated *Npc1*[-/-] mice with a single intraperitoneal (i.p.) administration of 2-hydroxypropyl-β-cyclodextrin (HPβCD) at P7. HPβCD mobilizes stored cholesterol in NPC1 deficient cells, and a single i.p. administration of HPβCD during development significantly extends the lifespan of NPC1 deficient mice[62,63]. Brains from treated mice were collected at P16 and assessed for myelination. Immunohistochemical imaging showed a widespread increase in MBP staining in *Npc1*[-/-] mice treated with HPβCD compared to those receiving vehicle alone (Fig. 7A). Additionally, HPβCD significantly rescued the loss of OLIG2-positive cells in the corpus callosum of *Npc1*[-/-] mice, suggesting improved cell survival (Fig. 7B). MBP, OLIG2, and SOX10 protein levels increased in Niemann–Pick C mouse forebrains treated with HPβCD compared to those treated with vehicle (Fig. 7C), further supporting the conclusion that HPβCD improved oligodendrocyte cell lineage survival, maturation, and myelination. Correspondingly, qRT-PCR showed that HPβCD partially restored the expression of myelin genes in the forebrain, indicative of an increase in cell differentiation (Supplementary Fig. 7A). Interestingly, improvements in myelination occurred without increased expression of OPC and immature oligodendrocyte marker genes (Supplementary Fig. 7A).

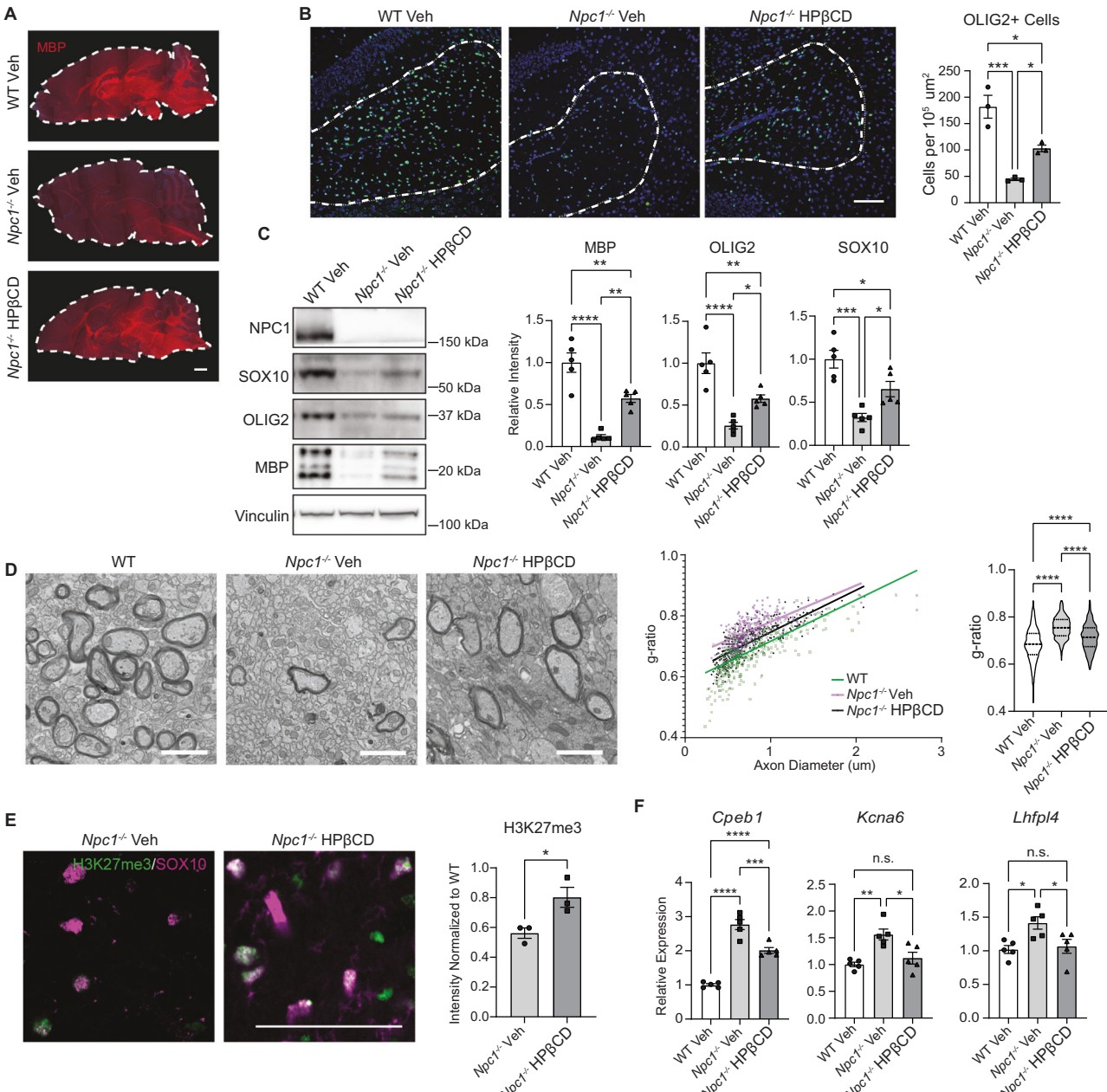

**Fig. 7 | A single injection of 2-hydroxypropyl-β-cyclodextrin rescues myelination defects.** Mice were administered 2-hydroxypropyl-β-cyclodextrin (4000 mg/kg, i.p.) (HPβCD) or a vehicle control (saline) (Veh) at P7. Brains were collected and myelination assessed at P16. Dashed line in A indicates the outline of the brain section. Midline sagittal sections were stained for (**A**) MBP (red) and (**B**) OLIG2 (green). Nuclei stained with DAPI (blue). The number of OLIG2+ cells in the corpus callosum (dashed line in **B**) quantified at right. **C** Western blot for SOX10, OLIG2, and MBP from forebrain lysates. Values were normalized to vinculin and relative to WT. Quantified at right. **D** TEM images of the corpus callosum. G-ratios were quantified (right). **E** Midline sagittal sections were stained for H3K27me3 (green) and SOX10 (magenta). Images were captured from the corpus callosum, and the intensity of H3K27me3 within SOX10+ nuclei was quantified (right). **F** Expression of neuronal genes in O4+ cells purified from P16 mice as determined by qRT-PCR. **A** Scale bars = 1000 μm; (**B**) 100 μm; (**D**) 2 μm; (**E**) 100 μm. **A** Experiment was performed on 3 biological replicates per group, all with similar results; dashed lines outline the sagittal brain. **D** Dashed lines in the violin plot indicate quartiles. Data are mean ± SEM. n.s. not significant, *$p < 0.05$, **$p < 0.005$, ***$p < 0.001$, ****$p < 0.0001$ by one-way ANOVA with Tukey post hoc test and multiple comparisons (**B–D**, **F**) or two-tailed unpaired Student's *t*-test (**E**). **B** $n = 3$ mice; **F** = 27.65; df = 2; $p = 0.0009$ (**C**) $n = 5$ mice; F = 35.97, 22.49, 16.55; df = 2; $p = <0.0001, <0.0001, 0.0004$ (**D**) Linear regression slopes: F = 1.574; df = 2; $p = 0.2078$; Linear regression intercepts: F = 256.9; df = 2; $p < 0.0001$; ANOVA: $n = 443, 301, 327$ axons (from 3 mice per group); F = 128.1; df = 2; $p = <0.0001$ (**E**) $n = 3$ mice; $t = 3.167$; df = 4; $p = 0.0340$ (**F**) $n = 5$ mice; F = 78.24, 10.83, 6.170; df = 2; $p = <0.0001, 0.0021, 0.0144$. Source data are provided as a Source Data file.

Analysis of electron micrographs of the corpus callosum showed that the myelin g-ratio in *Npc1*$^{-/-}$ mice was significantly increased compared to WT, signifying thinner myelin sheaths (Fig. 7D). This defect was partially rescued by HPβCD (Fig. 7D). The density of myelinated axons was also significantly diminished in the *Npc1*$^{-/-}$ corpus callosum compared to WT (Fig. 7D and Supplementary Fig. 7B). Although there was a

trend toward the rescue of this metric by HPβCD, statistical significance was not reached (Supplementary Fig. 7B). Notably, treatment with HPβCD significantly increased H3K27me3 staining of SOX10-positive cells in the corpus callosum (Fig. 7E). This was associated with the rescue of neuronal genes that were found to be epigenetically dysregulated in *Npc1*$^{-/-}$ oligodendrocyte lineage cells (Fig. 7F).

Together, these data demonstrate that correcting cholesterol homeostasis is sufficient to improve oligodendrocyte lineage epigenetic regulation, differentiation, and myelination in *Npc1*[−/−] mice.

### H3K27me3 defects occur in a cell-autonomous manner

Because hypomyelination occurs after *Npc1* deletion in either oligodendrocytes or neurons[30], we sought to determine if the deficits characterized above occur due to cell-autonomous or non-autonomous pathways. To accomplish this, we used mice with a conditional null allele of *Npc1*[64]. *Npc1*[flox/flox] mice were crossed with mice heterozygous for an *Npc1* null allele (*Npc1*[+/−]) and expressing Cre recombinase under the control of the *Olig2* (oligodendrocyte lineage-specific) or *Syn1* (neuron-specific) promoter. This cross generated experimental (*Cre+; Npc1*[flox/−]) and littermate control mice (*Cre+; Npc1*[flox/+]). Both *Olig2-Cre; Npc1*[flox/−] and *Syn1-Cre; Npc1*[flox/−] mice had reduced density of OLIG2-positive cells in the corpus callosum compared to controls (Supplementary Fig. 8), consistent with established myelination defects.

We next explored if the deficit in H3K27me3 resulted from cell-intrinsic defects or defective neuron-glial interaction. Midline brain sections from P16 mice were co-stained for H3K27me3 and SOX10. SOX10-positive cells in the corpus callosum of *Olig2-Cre; Npc1*[flox/−] mice had significantly reduced levels of H3K27me3 compared to control mice (Fig. 8A). In contrast, there was no difference in H3K27me3 staining intensity in SOX10-positive cells from *Syn1-Cre; Npc1*[flox/−] mice compared to controls (Fig. 8B), indicating that this epigenetic defect occurred in a cell-autonomous manner.

Finally, we treated *Olig2-Cre; Npc1*[flox/−] mice with HPβCD at P7 to determine if HPβCD acts directly on oligodendrocytes to improve H3K27me3. At P16, *Olig2-Cre; Npc1*[flox/−] mice treated with HPβCD demonstrated rescue of MBP staining (Fig. 8C), increased OLIG2-positive cells in the corpus callosum (Fig. 8D), and increased intensity of H3K27me3 staining in SOX10-positive cells compared to vehicle-treated controls (Fig. 8E). Together, these data show that HPβCD acts directly on oligodendrocytes to correct H3K27me3 deficits.

## Discussion

Here, we identify cell types that are affected early in the *Npc1*[−/−] mouse brain and gain insights into the pathogenesis of Niemann–Pick Disease type C. We demonstrate striking changes in oligodendrocyte lineage cells that appear as animals age from P6–P16, a period of developmental myelination. While the *Npc1*[−/−] oligodendrocyte lineage appears similar to WT at P6, a timepoint that precedes myelin formation, significantly reduced numbers of OLIG2-positive cells are present by P9 as a result of cell death. We suggest that this cell death is the result of stalled differentiation and maturation, as previous studies have shown that during development, oligodendrocytes that do not form myelin undergo cell death[38]. Defects in the oligodendrocyte lineage culminate at P16 in a phenotype characterized by robust alterations in gene expression, diminished numbers of immature and myelinating oligodendrocytes, and a loss of myelin proteins. These data indicate an essential role for the cholesterol transport protein NPC1 in normal oligodendrocyte lineage maturation, observations that complement prior studies demonstrating the importance of cholesterol bioavailability as rate-limiting for developmental myelination[12,13]. However, how cholesterol impacts oligodendrocyte differentiation and myelination has remained poorly understood.

Unanticipated insights into this process and the role of NPC1 in oligodendrocyte maturation come from the analysis of snRNA-seq data, which demonstrate the significant upregulation of genes associated with neurogenesis, ion transport, and synapse formation in *Npc1*[−/−] oligodendrocyte lineage cells. These genes are typically silenced as a requisite step in oligodendrocyte lineage maturation by the repressive epigenetic marks H3K9me3 and H3K27me3[44]. These marks are selectively lost in the *Npc1*[−/−] oligodendrocyte lineage but not in other cell types. Here, we have focused on the deficit in H3K27me3, as it appeared prior to the reduction in H3K9me3 and may be upstream. The transcriptional abnormalities we detected are seen in OPCs and immature oligodendrocytes as well as myelinating oligodendrocytes. This is noteworthy as recent studies demonstrate that cholesterol homeostasis plays an important role beyond oligodendrocyte development and impacts myelin maintenance into adulthood[65]. Thus, Niemann–Pick Type C provides a notable example of a developmental and degenerative nervous system disorder characterized by histone modification defects specific to the oligodendrocyte lineage that impact myelination.

We previously demonstrated that deletion of *Npc1* in either oligodendrocytes or neurons leads to dysmyelination, suggesting cell-autonomous and non-autonomous pathways contribute to the myelin phenotype[30]. Based on data presented here, we conclude that the H3K27me3 defect occurs in a cell-autonomous fashion, as the phenotype is present in *Olig2-Cre; Npc1*[flox/−] mice, but not in *Syn1-Cre; Npc1*[flox/−] mice. Furthermore, primary OPCs deficient in NPC1 recapitulate this maturation defect in vitro in the absence of neurons, and treatment with the histone demethylase inhibitor GSK-J4 rescues this phenotype. Notably, diminished H3K27me3 tightly correlates with the observed decrease in expression of the histone methyltransferase EZH2.

In NPC1-deficient mice, hypomyelination is severe in the forebrain but is more modest in the cerebellum, brainstem, and spinal cord[30–32], suggesting that oligodendrocytes that originate from different progenitor cell pools may have varying susceptibility to a lack of exogenous cholesterol. This pattern is also seen in our data, where diminished oligodendrocyte numbers and reduced H3K27me3 are seen in the corpus callosum but not in the cerebellar white matter. Interestingly, it was recently shown that cholesterol biosynthesis is one of the main pathways distinguishing oligodendrocyte lineage cells that originate in the forebrain from those that originate in the spinal cord[66]. The cells in the forebrain have lower cholesterol biosynthetic enzyme expression than cells in the spinal cord. This is coupled with increased expression of proteins associated with cholesterol endocytosis in forebrain cells[66]. These differences in cholesterol homeostasis may contribute to the sensitivity of forebrain oligodendrocyte lineage cells to NPC1 deficiency.

Our experimental evidence supports a model in which defective intracellular cholesterol trafficking underlies impaired oligodendrocyte lineage maturation due to NPC1 deficiency. The role of cholesterol in this process is suggested by data showing that *Npc1*[−/−] but not *Smpd1*[−/−] mice exhibit dysmyelination at P16. Further evidence comes from experiments with the cholesterol-mobilizing agent HPβCD. Partial rescue of myelination defects in *Npc1*[−/−] mice is achieved by a single i.p. administration of HPβCD at P7. While this treatment is known to significantly extend the lifespan of *Npc1*[−/−] mice, the mechanism underlying this effect has been incompletely understood. We argue that the rescue of oligodendrocyte maturation significantly contributes to this beneficial effect. Treatment of *Olig2-Cre; Npc1*[flox/−] mice with HPβCD leads to improved H3K27me3 and myelination, suggesting that HPβCD acts directly on oligodendrocytes. Notably, previously published data demonstrate that *Npc1*[−/−] mice receiving a single i.p. administration of HPβCD complexed with allopregnanolone at P7 live markedly longer than mice injected just 3 days later at P10[67]. This differential treatment efficacy during a narrow developmental window correlates with the timing of the peak oligodendrocyte lineage cell death in *Npc1*[−/−] mice demonstrated here.

Several mechanisms may link intracellular cholesterol trafficking defects to epigenetic dysregulation. In addition to its buildup in the lysosome, cholesterol accumulates in the mitochondria of NPC1 deficient cells, causing mitochondrial dysfunction and impaired energy metabolism[68–72]. The TCA metabolite α-ketoglutarate serves as a cofactor for Jumanji family of histone lysine demethylases, providing a

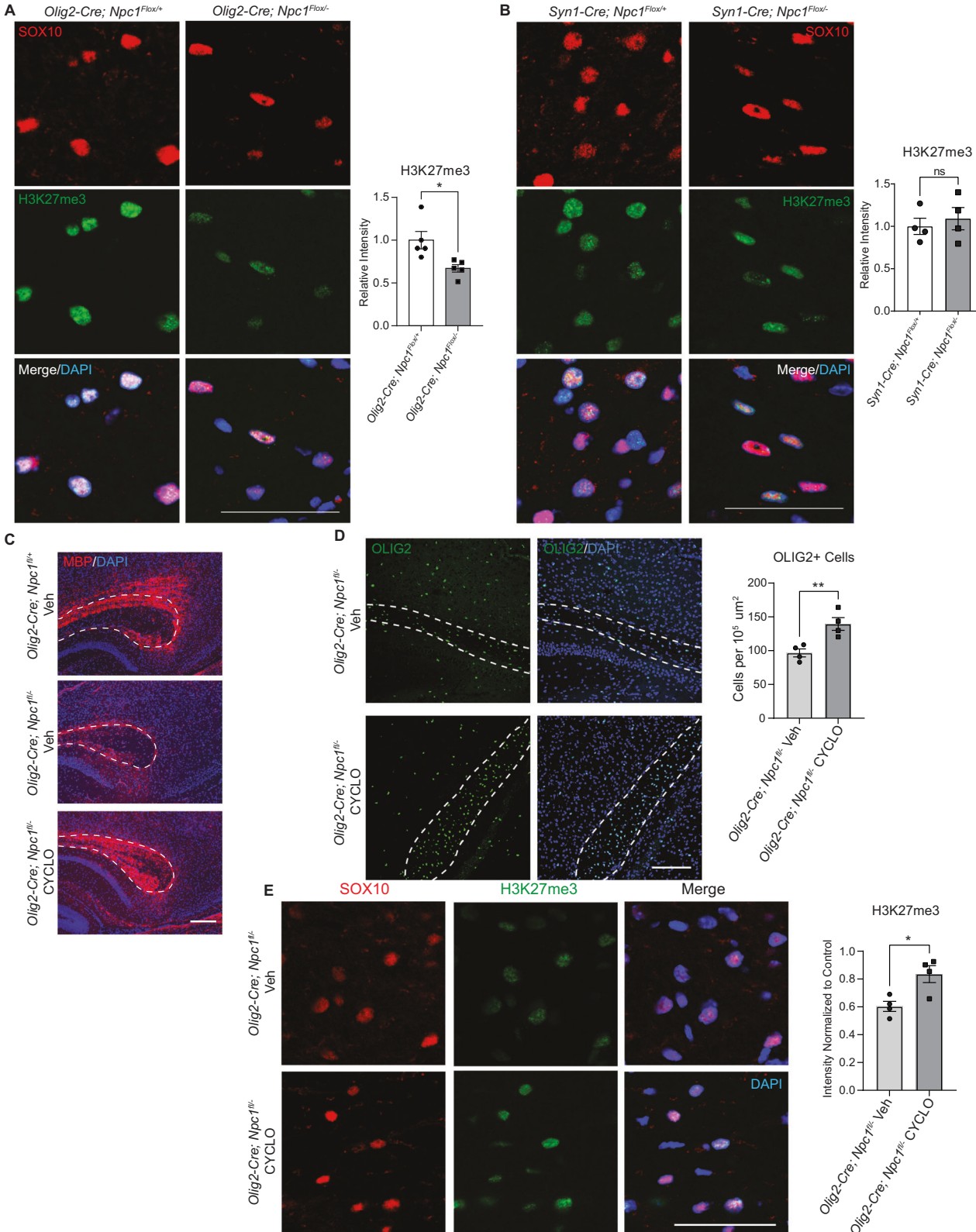

link between metabolism and epigenetic regulation[73]. Additionally, acetyl-CoA, which is decreased in the brains of $Npc1^{-/-}$ mice, acts as an enhancer of histone deacetylase activity[69,74–76] to influence gene expression. This mitochondrial dysfunction may also contribute to increased production of reactive oxygen species, as signs of increased oxidative stress have been reported in tissues from NPC1-deficient mice, as well as in human patients[72,77]. Additionally, cholesterol

bioavailability has been shown to affect signaling cascades that influence myelination and oligodendrocyte maturation, including mTORC1 and TFEB activity as well as kinase cascades[78–83]; iPathway analysis of differentially expressed genes in purified O4+ cells nominates several of these as potential contributors to the phenotype of $Npc1^{-/-}$ mice (Supplementary Fig. 9). Future studies will be aimed at linking alterations in these pathways with the epigenetic changes and disruptions of

**Fig. 8 | Cell-autonomous defects in H3K27me3 are corrected by 2-hydroxypropyl-β-cyclodextrin.** H3K27me3 staining (green) was performed on sagittal brain sections from (**A**) *Olig2-Cre;Npc1*[fl/+] and *Olig2-Cre;Npc1*[fl/−] or (**B**) *Syn1-Cre;Npc1*[fl/+] and *Syn-Cre;Npc1*[fl/−] mice at P16. Sections were co-stained for SOX10 (red) and DAPI (blue). Images were taken from the corpus callosum, and H3K27me3 intensity within SOX10 + cells was quantified relative to control. **C–E** *Olig2-Cre;Npc1*[fl/+] and *Olig2-Cre;Npc1*[fl/−] mice were administered 2-hydroxypropyl-β-cyclodextrin (4000 mg/kg, i.p.) (HPβCD) or a vehicle control (saline) (Veh) at P7. Brains were collected at P16. Sagittal sections were stained for (**C**) MBP (red), (**D**) OLIG2 (green), and (**E**) H3K27me3 (green). DAPI is in blue. Dashed lines outline the corpus callosum.

The experiment in (**C**) was done on 3 biological replicates per group, all with similar results. OLIG2+ cells in the corpus callosum were quantified normalized to area (**D**). H3K27me3 intensity in SOX10+ (red) cells of the corpus callosum were quantified relative to *Olig2-Cre;Npc1*[fl/+] mice treated with vehicle (**E**). Scale bars = 50 μm (**A**, **B**, **E**), 200 μm (**C**), and 150 μm (**D**). Data are mean ± SEM. n.s. not significant, *$p < 0.05$, **$p < 0.005$ by two-tailed unpaired Student's *t*-test. **A** $n = 5$ mice; $t = 2.968$; df = 8; $p = 0.0179$; (**B**) $n = 4$ mice; $t = 0.5540$; df = 6; $p = 0.5996$ (**D**) $n = 4$ mice; $t = 3.826$; df = 6; $p = 0.0087$; (**E**) $n = 4$ mice; $t = 3.274$; df = 6; $p = 0.0170$. Source data are provided as a Source Data file.

oligodendrocyte maturation that characterize the Niemann–Pick brain.

In summary, our findings identify an important role for NPC1 in oligodendrocyte lineage maturation and developmental myelination that is manifested through the epigenetic cascade required for this process. Moreover, we demonstrate unexpected impacts of Niemann–Pick C therapeutics on this pathway and suggest that this rescue contributes to beneficial effects in *Npc1*[−/−] mice. Our data broaden the target cells in the Niemann–Pick brain that contribute to disease phenotypes and that are amenable to rescue, and they provide insights into mechanisms by which intracellular cholesterol bioavailability regulates myelination.

## Methods

### Antibodies

**Primary antibodies.** The following primary antibodies (antigen [clone], dilution, vendor, cat. no.) were used for these studies: NPC1 [EPR5209], 1:500, Abcam, ab134113; NeuN [A60], 1:500, MilliporeSigma, MAB377; Ki67 [polyclonal], 1:200, Abcam, ab15580; Vinculin [hVIN-1], 1:2000, MilliporeSigma, V9131; Neurofilament [N52], 1:500, MilliporeSigma, MAB5266; H3K27me3 [C36B11], 1:200 [IF], 1:1000 [WB], Cell Signaling Technologies, 9733 S; H3K9me3 [D4W1U], 1:200 [IF], 1:1000 [WB], Cell Signaling Technologies, 13969 S; Histone H3 [96C10], 1:2000, Cell Signaling Technologies, 3638 S; MBP[12], 1:100 [IF], 1:500 [WB], Abcam, ab7349; OLIG2 [polyclonal], 1:500 [IF], 1:1000 [WB], MilliporeSigma, AB9610; SOX10 [SP267], 1:400, Abcam, ab227680; SOX10 [A-2], 1:200, Santa Cruz Biotechnology, sc-365692; H3K27me3 [polyclonal], ActiveMotif, cat. #39155; H3K27ac [polyclonal], ActiveMotif, cat. #39133.

**Secondary antibodies.** The following secondary antibodies (antigen, dilution, vendor, cat. no.) were used: goat anti-rabbit IgG (H + L)-HRP conjugate, 1:2000, Bio-Rad, 1706515; goat anti-mouse IgG (H + L)-HRP conjugate, 1:2000, Bio-Rad, 1706516; anti-rat IgG HRP conjugated, 1:2000, R&D Systems, HAF005; Alexa Fluor 488 goat anti-rat IgG (H + L), 1:500, Invitrogen, A11006; Alexa Fluor 594 goat anti-rat IgG (H + L), 1:500, Invitrogen, A11007; Alexa Fluor 594 goat anti-mouse IgG (H + L), 1:500, Invitrogen, A11032; Alexa Fluor 488 goat anti-rabbit IgG (H + L), 1:500, Invitrogen, A11008.

### Mice

*Npc1*[nih] BALB/cJ mice were obtained from Jackson Laboratories (#003092) and backcrossed to C57BL6/J for at least 10 generations. *Npc1*[−/−] mice were generated as F1 hybrids by crossing *Npc1*[+/−] mice on BALB/cJ and C57BL/6J backgrounds. This hybrid cross allowed for the restoration of Mendelian frequency of *Npc1*[−/−] mice[84]. Genotyping was performed as previously described[64]. *Smpd1*[−/−] mice were maintained on the C57BL6/J background. Genotyping was done as described by ref. [58]. Floxed-*Npc1*[64], *Syn1-Cre* (003966, Jackson Labs), and *Olig2-Cre* (025567, Jackson Labs) mice were maintained on the C57BL6/J background. Male *Npc1*[flox/flox] mice were crossed with female *Cre*+; *Npc1*[+/−] mice to generate experimental groups. Lox/Cre mice were genotyped as previously described[64]. Littermate mice were genotyped prior to use in experiments. Approximately equal numbers of male and female mice were used in these studies. All animal procedures were approved by the University of Michigan Committee on the Use and Care of Animals (PRO00010017). Mice were housed in a 12-h dark/light cycle, in a facility with an ambient temperature ranging from 20.6–23.9 °C and a humidity between 30–70%.

### Purification of mouse primary OPCs by magnetic-activated cell separation (MACS)

OPCs were purified from whole mouse brains at P6 (for cell culture) or P16 (for histone extraction, western blot, and genetic analyses). OPC isolation was performed using reagents from Miltenyi Biotec (Auburn, CA) following manufacturer instructions. Briefly, mice were deeply anesthetized using ice (P6 mice) or isoflurane (P16) and their brains were removed. Brains were dissociated into single cell suspensions using the Neural Tissue Dissociation Kit (cat. 130-092-628) and gentleMACS Octo Dissociator (130-095-937). Dissociated cells were passed through a 70um cell strainer, then washed and resuspended in MACS buffer (0.5% bovine serum albumin in phosphate-buffered saline). Cells were incubated on ice with an Fc blocking reagent (130-092-575) for 10 min, followed by microbeads targeted to O4 (130-094-543) or PDGFRα (130-101-502) for 15 min. Cells were then run through LS columns (130-042-401) and washed 3X with MACS buffer. The flow through cells were pelleted and flash frozen. Purified OPCs were flushed from the column with a MACS buffer. OPCs were counted, pelleted, and either flash frozen or resuspended in media and plated for culture.

### OPC culture and differentiation

Mice were anesthetized on ice and decapitated. Primary OPCs were isolated from P6 mouse brains by MACS using PDGFRα microbeads as described above. Cells were plated onto 15 mm PDL-coated cover glass at a density of $5 \times 10^4$ cells per cover glass. Cells were plated in OPC media (50 μg/mL insulin, 1% N2 supplement, 2% B27 supplement, 1% pen/strep, 0.1% BSA, 40 ng/mL FGF2, 20 ng/mL PDGF-AA in DMEM-F12) and allowed to settle on the cover glass for 3 h and media was then replaced with fresh OPC medium. After 24 h, OPC media was replaced with differentiation medium (50 μg/mL insulin, 1% N2 supplement, 2% B27 supplement, 1% pen/strep, 40 ng/mL triiodothyronine (T3) in DMEM-F12) containing either GSK-J4 (0.5 μM), GSK-J5 (0.5 μM), or DMSO. Differentiation medium was replaced every 2 days. Cells were fixed and stained after 4 days in differentiation medium.

### Acid extraction of histones

Histones were extracted as described by ref. [85]. Briefly, frozen cell pellets were lysed in hypotonic lysis buffer (10 mM Tris HCl pH8.0, 1 mM KCl, and 1.5 mM MgCl2, protease inhibitor and phosphatase inhibitor, Sigma-Aldrich) for 30 min. Sulfuric acid (0.4 N) was added, and samples were incubated overnight at 4 °C with rotation. Samples were centrifuged at 16,000 g for 10 min, supernatants were collected, and proteins were precipitated in 33% trichloroacetic acid. Histones were washed with acetone and resuspended in deionized water. Protein concentrations were measured with a Pierce™ BCA Protein Assay Kit (Thermo Fisher 23227).

## Western blot

Mice were anesthetized with isoflurane and perfused with saline before tissues were collected and flash frozen in liquid nitrogen. Tissues were homogenized and sonicated in RIPA buffer containing HALT protease-phosphatase inhibitor (Thermo Fisher 78430) and 0.625 mg/mL N-ethylmaleimide (Sigma E3876). Protein concentrations were determined by DC-protein assay (BioRad) and normalized. Proteins were separated on NuPAGE 4–12% Bis-Tris Protein Gels (Thermo Fisher NP0336BOX) and transferred to Immobilon-P PVDF (0.45-μm pore size, Merck Millipore). Immunoreactivity was detected with Super-Signal West Pico PLUS Chemiluminescent Substrate (Thermo Fisher) and an iBright (Thermo Fisher). Images were quantified on ImageJ, with band intensity being normalized to the indicated loading control.

## Immunocytochemistry

Cells were washed three times with PBS and fixed with 4% PFA for 20 min at room temperature. Cells were washed 3X with PBS, permeabilized with 0.1% triton for 20 min, and then placed in blocking buffer (10% goat serum, 1% BSA in PBS) for 1 h. Cells were incubated with primary antibody in blocking buffer overnight at 4 °C. Slides were washed 3X with PBS and incubated with secondary antibody diluted in blocking buffer for 1 h at room temperature. Slides were washed 3X with PBS and mounted using Vectashield plus DAPI (Vector Laboratories).

## Immunohistochemistry

For tissue preparation, mice were anesthetized with isoflurane and perfused with saline followed by 4% PFA. Tissues were isolated and post-fixed in 4% PFA overnight at 4 °C. Tissues were transferred to 30% sucrose at 4 °C for 48 h before being frozen in O.C.T. (Tissue-Trek). Sections were cut at 10 μm using a Leica CM1900 cryostat and adhered to slides. Slides were incubated in a solution of 0.1% Triton, 10% goat serum, and 1% BSA for 30 min. Slides were placed in blocking buffer (10% goat serum, 1% BSA in PBS) for an additional 30 min before incubating in primary antibody diluted in blocking buffer overnight at 4 °C. For histone stains, blocking buffer included the addition of 0.1% Triton. Slides were washed 3X in PBS and incubated with secondary antibodies diluted in a blocking buffer for 1 h at room temperature. Slides were then washed 3X with PBS and mounted with Vectashield plus DAPI (Vector Laboratories)

## Microscopy

Confocal images were captured using a Nikon N-SIM + A1R microscope. Three images were quantified and averaged for each sample. For histone methylation quantifications, the intensity of histone methylation staining was only measured in cells staining positive for SOX10 or NeuN. The intensity for all cells across 3 images were averaged for each biological replicate. All quantifications were done using an unbiased pipeline in CellProfiler Analyst Software. Whole brain sagittal sections were captured and stitched together using a Zeiss Axio Imager Z1 microscope with an automated stage.

## ChIP-seq

ChIP-seq was performed on O4-microbead-purified cells from P16 WT and $Npc1^{-/-}$ mice. Mice were anesthetized with isoflurane and decapitated. Cells were purified by MACS as described above. Cells were collected from six mice of each genotype and pooled together for one ChIP-seq run. Chromatin preparation and ChIP-Seq analysis were performed by Active Motif, Inc. using 4 μg the antibodies against H3K27me3 (ActiveMotif, cat. #39155) and H3K27ac (ActiveMotif, cat. #39133). For H3K27me3 samples, *Drosophila melanogaster* DNA was added as a minor fraction to standardize reactions using an antibody recognizing the *Drosophila*-specific histone variant H2Av.

The 75-nt sequence reads generated by Illumina sequencing were mapped to the mouse genome using the BWA algorithm with default settings. Only reads that passed the Illumina purity filter, aligned with no more than two mismatches, and mapped uniquely to the genome were used in subsequent analysis. Duplicate reads were also removed. For H3K27me3 samples, the number of test tags were normalized by the same number of *Drosophila* tags for each sample. For H3K27ac samples, the tag number of all samples was normalized to the number of tags present in the smallest sample. Alignments were extended in silico at their 3′-ends to a length of 200 bp, which corresponds to the average fragment length in the size-selected library, and assigned to 32-nt bins along the genome. The resulting data was stored in bigWig files. MACS2[86] and SICER[87] were used to call peaks, and average profile for tag distributions were generated using ngsplot v2.61. Integrated Genomics Viewer (IGV, Broad Institute) was used to display H3K27me3 and H3K27ac traces from pooled bigWig files.

## Bulk RNA-seq

RNA was collected from O4-microbead-purified cells using RNeasy Micro Kit (Qiagen 74004) following the manufacturer's instructions. cDNA was prepared from 3 biological replicates from each group using NEBNext Poly(A) mRNA Magnetic Isolation Module (NEB, E7490) and xGen Broad-range RNA Library Prep (IDT, 1009813) with xGen Normalase UDI Primers (IDT). This pool was subjected to 151 bp paired-end sequencing according to the manufacturer's protocol (Illumina Nova-Seq). BCL Convert Conversion Software v3.9.3 (Illumina) was used to generate de-multiplexed Fastq files. Raw reads were trimmed with bbduk[88], and the quality of the trimmed reads was evaluated with FastQC v0.11.9[89] and MultiQC v1.12[85]. The trimmed reads were then mapped to the mouse reference genome (GRCm38) using the STAR aligner v2.7.9a[90], and the quality of the alignments was evaluated with MultiQC. The alignment options were adjusted to improve the percentage of reads mapping uniquely to the reference. Next, featureCounts[91] was used to assign count estimates to the genes, and MultiQC was used again to evaluate the quality of the resulting gene by sample count matrix. The dimensionality of the count matrix was reduced to two principal components with Principal Components Analysis (PCA)[92].

Before running Differential Expression (DE) analysis, genes that had less than ten counts across all samples were removed. DE analysis was then performed with the DESeq2 library[93] with default settings in R to determine differentially expressed genes, with the condition variable as the main factor within the design formula. The results were filtered to obtain the differentially expressed genes with an adjusted $p < 0.05$ and $|\log_2(\text{fold-change})| \geq 0.1$. In addition to the results table, volcano plots were constructed to quickly identify statistically significant genes with a large magnitude of change between conditions. Gene Ontology (GO) enrichment analysis was run on DE genes using Fisher's exact test from the topGO library[94] in R. iPathway analysis was performed by Advaita Corporation.

## Single-nucleus RNA-seq

Mice were euthanized with isoflurane followed by decapitation. Brains were collected from three P16 mice from each genotype (WT and $Npc1^{-/-}$), and the forebrain was dissected and flash frozen in liquid nitrogen. Nuclei were isolated using a protocol adapted from ref. [95]. Forebrains were homogenized using a glass Dounce tissue grinder (25X with pestle A and 25X with pestle B) in 2 mL of EZ PREP buffer (Sigma NUC-101). Homogenized tissues were incubated on ice for 5 min with an additional 2 mL of EZ PREP. Nuclei were pelleted by centrifuging at 500 g for 5 min at 4 °C. Nuclei were washed with 4 mL EZ PREP and incubated on ice for an additional 5 min. Nuclei were centrifuged again and washed in 4 mL of nuclear suspension buffer (0.01% BSA, RNase inhibitor [Takara 2313 A]), followed by centrifugation for 5 min at 500 g. The nuclear pellet was resuspended in 1 mL of nuclear suspension buffer before filtering through a 35 μM cell strainer (Corning 352235) and counted. Nuclei were diluted to a final concentration of 800–1500 cells/μL. Single-nucleus libraries were

generated using the 10X Chromium Controller system. Libraries were sequenced using an Illumina NovaSeq 6000.

The STARsolo aligner v2.7.9a[96] was used to generate an index to the mouse reference genome (GRCm38) and to align the fastq files from each sample. The quality of the alignments was assessed with MultiQC v1.12[89]. After alignment, the following cell filters were applied to the resulting count matrix for each sample. Cells with minimal total UMI counts were removed with the EmptyDrops methodology[97] and the nExpectedCells parameter was adjusted to 10,000 cells to increase the total number of cells retained per sample. Multiplets were identified and removed with DoubletFinder[98] using the default parameters. The percentage of reads within each sample aligning to mitochondrial-encoded genes was examined employing violin plots with the ggplot2 library version 3.3.6[99] in R, and a consistent threshold of 0.10 was identified across samples to remove cells exhibiting high mitochondrial gene expression (Supplementary Fig. 10). Finally, genes expressing less than a total of ten UMI counts across all cells were removed. Density plots for the total UMI counts and total unique gene counts per sample were constructed to examine the effects of cell filtering (Supplementary Fig. 10).

After applying all cell and gene quality control filters, the Adaptively-thresholded Low-Rank Approximation (ALRA) algorithm[100] was utilized to impute dropout events. After imputation, the Seurat v4.1.0 library[101] in R was employed to run log-normalization on the matrix with a default scale factor of 10,000. The top 3000 variable features were then identified utilizing the FindVariableFeatures function in Seurat with default parameters. The ScaleData function in Seurat was used to center each gene's expression values by subtracting the average expression and scale the gene's expression values by dividing the centered values by their standard deviations. Both Principal Component Analysis (PCA)[92] and Uniform-Manifold Approximation Projection (UMAP)[102] dimensionality reduction techniques, with varying parameter specifications, were employed using the RunPCA and RunUMAP functions in Seurat before clustering. A shared nearest neighbor (SNN) graph was constructed with differing numbers of principal components using the FindNeighbors function. The FindClusters function was run to identify cell clusters with varying specifications for the resolution parameter. The cell clusters were then visualized with scatterplots for $Npc1^{-/-}$ and WT samples separately using the UMAP projections and ggplot2 library.

Differential expression (DE) analysis was then performed with the Seurat FindMarkers function, which employs a Wilcoxon rank-sum test[103] iteratively between each cell cluster and all other cells. A dot plot was created with the ggplot2 library to view the normalized and scaled expression of previously determined cell type marker genes[34–36,43] and the top three DE marker genes within each cell cluster (Fig. 1B). A heatmap was also created with ggplot2 to view the expression of the marker genes within each cell cluster (Supplementary Fig. 1). Cell type labels were assigned manually to each cluster based on the exploration of the results from the DE analysis and dot plot visualization. Following cell typing, the FindMarkers function was applied again, with the Wilcoxon rank-sum test, to run a DE analysis between $Npc1^{-/-}$ and WT samples within each cell type independently. Genes within each cell type were identified as being differentially expressed between conditions if they had an adjusted $p < 0.05$ and |log$_2$(fold-change)| ≥0.5. Gene Ontology (GO) enrichment analysis was run on DE genes using Fisher's exact test from the topGO library[94] in R. Differential Abundance (DA) analysis was performed for each cell type cluster between the WT and Npc1$^{-/-}$ samples using the propeller function[104] in R to identify statistically significant differences in the proportion of cell types between conditions.

## TUNEL assay
Paraffin-embedded tissues were cut on a Reichert-Jung 2030 microtome into 10 µm sections and placed on Thermo Scientific Superfrost Plus microscope slides. Sections were adhered onto slides in an oven at 55–60 °C for 1 h. Samples were deparaffinized and stained using a Click-iT™ Plus TUNEL Assay Kit (Thermo Fisher C10618) following the manufacturer's instructions. Slides were washed 3X with PBS and mounted with Vectashield plus DAPI (Vector Laboratories).

## qRT-PCR
RNA was collected from cells or tissues using TRIzol® (Thermo Fisher) following the manufacturer's instructions. RNA was converted to cDNA using the High Capacity Reverse Transcription kit (Applied Biosystems 4368814). Quantitative real-time PCR was performed in technical triplicates using 20 ng of cDNA. TaqMan™ probes (Thermo Fisher) targeting mouse *Sox10* (Mm00569909), *Olig2* (Mm01210556), *Mbp* (Mm01266402), *Smoc1* (Mm00491564), *Plp1* (Mm01297210), *Cspg4* (Mm00507257), Enpp6 (Mm00624107), *Dusp15* (Mm00521352), *Cpeb1* (Mm01314928_m1), *Kcna6* (Mm00496625_s1), *Lhfpl4* (Mm00625122_m1) and *Cpsf2* (Mm00489754) were used. RT-qPCR was conducted using an ABI7900HT Sequence Detection System and relative expression was calculated by the 2^(-ΔΔCT) method, with *Cpsf2* being used as a control for normalization.

## 2-hydroxypropyl-β-cyclodextrin treatment
P7 mice were given a single i.p. injection of 2-hydroxypropyl-β-cyclodextrin (20% in saline) at a dose of 4000 mg/kg as previously described[59,62]. Some mice received injections of saline alone as a vehicle control. Mice were anesthetized with isoflurane and perfused with saline alone (tissues used for western blot or qRT-PCR) or saline followed by 4% PFA (tissues used for immunohistochemistry). Brains were collected from injected mice at P16 for analysis.

## Electron microscopy
Mice were anesthetized with isoflurane and perfused with 0.9% normal saline followed by 2% paraformaldehyde and 2.5% glutaraldehyde in 0.1 M sodium cacodylate buffer. The corpus callosum was removed and post-fixed in perfusion solution overnight, followed by fixation in 1% osmium tetroxide solution for 1 h at room temperature. After dehydration, tissues were embedded in epoxy resin. For transmission electron microscopy, ultrathin sections were cut, and images were captured on a JEOL JEM-1400 Plus LaB6 TEM imaging system at 3000 magnification. ImageJ was used to measure the area within the inner (area$^{in}$) and outer (area$^{out}$) rims of each myelin sheath. Axon caliber was derived from the area$^{in}$, where caliber = 2r and area$^{in}$ = πr². G-ratios were calculated as area$^{in}$area$^{out}$. The g-ratios for at least 100 axons per sample were calculated.

## Statistical analysis
GraphPad Prism 9.0 was used to determine significance ($P < 0.05$), $F$ (F-statistic) and $t$ (T-statistic) values. Unpaired Student's $t$-test (2 tailed) and one-way or two-way ANOVA were used as indicated in the figure legends. A $P$ value less than 0.05 was considered significant. All error bars are +/− SEM.

## Reporting summary
Further information on research design is available in the Nature Portfolio Reporting Summary linked to this article.

## Data availability
The data and materials supporting the findings of this study are available from the corresponding authors upon request. The raw ChIP-seq, snRNA-seq, and bulk RNA-seq data have been deposited in the GEO database under the series accession code GSE221610. Sequence alignment was done using the mouse reference genome GRCm38 [https://www.ncbi.nlm.nih.gov/assembly/GCF_000001635.20/]. Source data are provided with this paper.

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

## Acknowledgements
This work was funded by the National Institutes of Health (R01 NS122746 to A.P.L.; T32 GM113900 to T.J.K.; R01 DA051913, R01 DA051908 to D.A.J.) and the Firefly Fund (to A.P.L.). We thank Mark Schultz, Siva Kumar Natarajan, Matthew Pun, and Sriram Venneti for their insightful discussions. We also thank Halle Kunkel for allowing us to use her illustrations of oligodendrocytes (Fig. 4A). Library preparation and next-generation sequencing for RNA-seq experiments were carried out by the University of Michigan Advanced Genomics Core.

This manuscript has been authored by UT-Battelle, LLC under Contract No. DE-AC05-00OR22725 with the U.S. Department of Energy. The United States Government retains and the publisher, by accepting the article for publication, acknowledges that the United States Government retains a non-exclusive, paid-up, irrevocable, world-wide license to publish or reproduce the published form of this manuscript, or allow others to do so, for United States Government purposes. The Department of Energy will provide public access to these results of federally sponsored research in accordance with the DOE Public Access Plan (http://energy.gov/downloads/doe-public-access-plan).

## Author contributions
Conceptualization: T.J.K., A.T., K.A.S., D.A.J., and A.P.L. Investigation: T.J.K., A.T., K.A.S., and J.M. Funding acquisition: T.J.K., D.A.J., and A.P.L. Critical reagent (Smpd1–/– mice): E.H.S. Supervision: D.A.J. and A.P.L. Writing – original draft: T.J.K., A.T., K.A.S., D.A.J., and A.P.L. Writing – review & editing: T.J.K., A.T., K.A.S., J.M., E.H.S., D.A.J., and A.P.L.

## Competing interests
The authors declare no competing interests.
