## [Peer Review File · Nature Communications]

REVIEWER COMMENTS

Reviewer #1 (Remarks to the Author):

In this study the authors have examined the effects of *Npc1* deficiency on oligodendrocyte lineage cells. Earlier studies from this group have shown that deletion of *Npc1* in oligodendrocytes causes neurodegeneration. In this study, single nuclei RNA-seq (snRNA-seq) of P16 *Npc1*^{-/-} cells from the forebrain revealed a reduction in immature and mature oligodendrocytes but not OPCs. This was verified by immunofluorescence analysis showing that severe reduction in oligodendrocytes occurred between P9 and P16 due to increased cell death. Intriguingly, in addition to reduction in transcripts involved in myelination and cholesterol biosynthesis, genes involved in neurogenesis and synapse formation were upregulated in the mutant immature oligodendrocytes. H3K27me3 histone modification as well as *Ezh2* were reduced in *Npc1*^{-/-} immature oligodendrocytes (and also in OPCs) *in vivo* and in isolated oligodendrocyte lineage cells. GSK-J4 inhibitor of H3K27 demethylases reversed the proportion of differentiated oligodendrocytes back to wild type levels. ChIP-seq for H3K27me3 and H3K27ac showed reduced H3K27me3 occupancy at the promoters of 16 upregulated genes related to neuronal projection or synapse formation, and of these 10 also had increased H3K27ac occupancy at the promoter. In the final set of experiments, that authors treated *Npc1*^{-/-} mice with a single injection of HPbetaCD at P7 to mobilize stored cholesterol, and they show that oligodendrocytes and myelin were restored, neuronal genes were repressed, and H3K27me3 was increased closer to wt levels.

The authors provide a very interesting set of findings pointing to epigenetic dysregulation in immature oligodendrocytes deficient for *Npc1*, which has important implications in the pathogenesis and pathophysiology of Niemann-Pick disease. They provide a robust set of data to demonstrate defects in repressive histone posttranslational modification in immature oligodendrocytes that are likely to be responsible for the observed upregulation of neuronal genes in mutant oligodendrocytes. The data are clearly presented with adequate statistical analyses, and the manuscript is logically organized. The following are some questions and comments that can be addressed to clarify some ambiguities and strengthen the main conclusions.

1. H3K27me3 is reduced in OPCs and immature O4+ oligodendrocytes in *Npc1*^{-/-} brains. However, there is not much phenotypic change in *Npc1*^{-/-} OPCs, despite detection of *Npc1* transcripts in OPCs in previously published databases at least at a level that is comparable to if not higher than that in immature or mature oligodendrocytes. Do the authors think that the changes in H3K27me3 are already affecting the function of OPCs even if they do not detect obvious changes in their number or proliferation?
2. If altered cholesterol trafficking is upstream of epigenetic changes, how does one explain the reduced H3K27me3 in OPCs? In Figure 7, mobilizing cholesterol systemically in the mutant mice restores the level of H3K27me3, suggesting that cholesterol mobilization defects are upstream of and lead to reduced H3K27me3. While cholesterol is present in all cells, it is not until the cells reach the immature oligodendrocyte stage that cholesterol biosynthesis becomes prominent and is critically needed for subsequent maturation into myelinating oligodendrocytes. How do the authors think H3K27me3 is downregulated in OPCs (i.e. is it due to altered cholesterol trafficking in OPCs or some other mechanism)?
3. Related to question #2, does treating PDGFRalpha-positive purified OPCs (not immature oligodendrocytes) with HPbetaCD or a similarly acting agent also restore H3K27me3 levels?
4. Were neuronal genes also upregulated in OPCs, in addition to immature oligodendrocytes?
5. I am trying to understand if all the observed epigenetic dysregulation and neuronal gene expression can be simply attributed to trafficking defects of cholesterol, i.e. altered level and location and function of cholesterol in the cells, or whether *Npc1* defects in OPCs can lead to alterations in major signaling pathways that are critical for the terminal differentiation of oligodendrocytes, rather than maturation defects directly due to altered cholesterol trafficking. The

authors discuss this a little in the Discussion when they allude to mTOR and TFEB signaling, but they mix up the function in different stages of oligodendrocyte lineage cells, making it difficult to decipher how they are interpreting the OPC data. This question is stemming from the paucity of data on the critical role for cholesterol in OPC-to-oligodendrocyte differentiation, compared to the well documented role for cholesterol in newly formed oligodendrocytes transitioning into mature myelinating oligodendrocytes. The data are not entirely consistent with the exclusive role of *Npc1*^{-/-} defect – cholesterol defect – H3K27me3 transcriptional repression in immature oligodendrocytes. H3K27me3 changes may already be present prior to the stage at which cholesterol is considered critical.

6. The neuronal genes whose upregulation in *Npc1*^{-/-} cells is reversed by HPβCD seem to be already expressed, albeit at low levels for some, in wt oligodendrocyte lineage cells. They may belong to a family of neuronal genes that are transcribed in OL lineage genes at low abundance but probably not at a functionally significant level. It is possible that upregulation of these genes occurs in *Npc1*^{-/-} because the chromatin is already somewhat accessible. Would the authors please provide information as to what proportion of the neuronal genes upregulated in *Npc1*^{-/-} immature oligodendrocytes are detected in wt cells and what proportion of the neuronal genes are completely transcriptionally silent/more tightly repressed in wt cells? It would be interesting to know if the epigenetic dysregulation only affects neuronal genes in oligodendrocytes that are incompletely repressed.

7. Would the authors speculate on whether or how their finding that specific neuronal genes are upregulated in *Npc1*^{-/-} immature oligodendrocytes could potentially be linked to the documented neuronal dysfunction in NP disease?

Overall, this is a very interesting manuscript that opens up many additional questions that could be explored experimentally.

Reviewer #2 (Remarks to the Author):

Kunkel and colleagues studied molecular links between oligodendrocyte development, cholesterol and Niemann-Pick Type C (NPC) disease. In this rare autosomal-recessive lysosomal storage disorder, defects in either NPC1 or NPC2 protein impair the exit of unesterified cholesterol from late endosomes/lysosomes and cause accumulation of this and other lipids within the organelle. At present, it is a veritable mystery how this well-described cellular defect provokes distinct forms of the disease with age of onsets ranging from perinatal to adult and symptoms ranging from perinatally lethal liver failure to neurologic/psychiatric decline progressing over years. *Npc1*-deficient mice recapitulate key visceral and neurologic symptoms and die prematurely. As shown previously by many groups including the authors, NPC1 deficiency impairs the development of oligodendrocytes (Yu & Lieberman, 2013; Takikita et al., 2004; De Nuccio et al., 2019), and leads to hypomyelination (e.g. Palmeri et al., 1994; Higashi et al., 1995; Walterfang et al., 2010) in animal models and patients. However, the mechanisms have remained unknown. Despite substantial advances, no curative therapy is currently available or under clinical development. Therefore, new mechanistic insight and new therapeutic targets are highly welcome. The work by Kunkel and colleagues has the potential to deliver on this: it suggests a new epigenetic mechanisms that impairs myelination in NPC1-deficient mice and that may serve as new drug target. Their work has the great merit to start by an unbiased examination of brain cells and their reaction to NPC1 deficiency using single-cell/nucleus transcriptome analyses. Surprisingly, only one publication so far reported NPC disease-related results obtained by this fertile approach (Cognoux et al., 2020), which makes the authors' data all the more interesting and valuable. The present work shows that the defect in oligodendrocyte development in NPC1-deficient mice is caused by decreased levels of histone methyltransferase EZH2, which is part of the polycomb repressor complex 2, and the transcriptional repressor H3K27me3. The involvement of H3K27me3

in oligodendrocyte development has been discovered within the last years (see Wang et al., 2020 Cell Rep; Bartosovic 2021 Nat Biotech; Zhang 2022 Nat Commun), but the evidence that this pathway is perturbed in NPC disease is new and unexpected. The authors show in vitro that pharmacologic inhibition of specific demethylases reverts the differentiation defect. And they suggest that the defect is specific to NPC1 deficiency and can be reverted in mice by treatment with beta-cyclodextrin.

The results presented by Kunkel and colleague are potentially of high significance for the Niemann-Pick field, and beyond, notably for diseases where defects in epigenetic regulation of brain cell differentiation may destabilize neuronal circuits and provoke their dysfunction. The methods are sound and described in sufficient detail except for instances indicated below. However, in its present form the manuscript is unsuitable for publication, and requires a major overhaul. Several conclusions are not supported by the data, and key questions need to be addressed experimentally to mark a significant advance. The analysis and presentation of data must be improved, and the organisation of the manuscript needs to be revised. The authors should consider the following points:

Major points

- The authors should revise their Abstract, Introduction and Discussion to explain more succinctly the overall goal of the study. In its present form, the declared aim seems to oscillate between "mechanisms of oligodendrocyte development" and "disease mechanisms in NPC". Notably the Introduction exposes fundamental aspects of oligodendrocyte development, although factually the work aims at disease mechanisms in NPC disease (see first paragraph of Results and Discussion), evidently with a focus on oligodendrocytes. In any case, the authors should also consider the following when revising the Introduction

- - Lines 61-68: The description of oligodendrocyte precursor cells needs to be revised and the reviews updated. The term "oligodendrocyte precursor cells" is often used in an ambiguous manner, designating precursors of oligodendrocytes, but also resident cells within the brain (NG2-positive), whose function is still debated. In any case, the field dealing with this cell type(s) has seen a revolution within the last decade, which should be acknowledged by the authors by citing more recent articles/reviews.

- - Lines 69-77: This paragraph should be revised, as it confounds several topics and it does not reflect the current state of knowledge. For example, the first statement concerning cholesterol and oligodendrocytes differentiation is unclear. Can cholesterol be considered a signal controlling oligodendrocytes differentiation? One gets the impression that the authors confound oligodendrocyte differentiation with myelin biogenesis. These are distinct developmental processes.

- The use of single cell/single nucleus analyses has become a method of choice to identify cell type-specific reactions to disease, and therefore the authors' data are highly welcome as new source for subsequent analyses in the field. However, the explanation and display of results including quality control data and differently expressed genes should be improved to follow established standards. Moreover, the data are not explored to reveal their full potential. Some suggestions:

- - The authors should show quality control-relevant data in main or supplementary figs. This includes histograms with the frequency distributions of cellular and gene UMI counts as well as mitochondrial gene counts together with the chosen cutoffs. They should further show the violin plots that are mentioned in line 558.

- - Lines 536-549: It is unclear whether single nuclei from individual animals were pooled or whether they were bar-coded to track interindividual variation and batch effects. This should be explained.

- - Lines 581-582: The authors should explain how marker genes of cell types were chosen. Reference(s) relevant to their phrase "previously determined" should be given.

- - Fig. 1A. The authors should show in Fig. 1 UMAP plots not only for wildtype but also for mutant mice to allow for visual comparison of cell composition. The axes should be labeled according to standards indicating the type of plot. The colors in the legend boxes are barely visible, the authors could increase the size of the boxes.

- - Fig. 1B: The authors should consider to replace or complement the bubble matrix by more informative plots displaying expression levels of selected marker genes in the different cell clusters. For example, violin plots reveal the distribution of values in each cluster in a more

accessible manner (color coding in bubbles is difficult for some readers).

- - Fig. 1C: The authors should expand the figure and show numbers of both up- (positive) and down-regulated (negative) genes separately.

- - Lines 126-129, Fig. 1D: The authors should represent the percentages of cells from all individual clusters in wildtype and mutant animals using stacked bar/column plots in order to provide a more detailed view on the changes across cells.

- Lines 129-167, Fig. 1D,E & Fig. 2, 3: These parts of the ms should be reorganized, as the sequence of results presented and their display in figures is confusing. The complementation/validation of snRNAseq data is an important aspect. On the other hand, as stated by the authors, their data confirm previously published results. Therefore, in these paragraphs and figures the authors should clearly indicate what has been shown in previous publications and what is new. Evidently, the focus should be on new results. To this end, experiments bringing new insight would be immunohistochemical staining of stage-specific markers to distinguish myelinating from non-myelinating oligodendrocytes. These markers should probably include those used to identify the different stages in their snRNA-derived cell clusters (see Fig. 1B, and PCR targets: lines 159-167; see also below). The choice of ages that were investigated should be explained. To further streamline this part, authors should move panel E from Fig. 1, where it is misplaced, to Fig 2. Further, the PCR experiments (Lines 159-167; Fig. 3A) performed on forebrain lysates stand isolated. The indicated rationale for the experiments "to further analyse the oligodendrocyte lineage" is rather weak and out of context. These results are part of the results presented in the paragraphs before and should be integrated after line 134. Accordingly panel A of fig. 3A should be moved to Fig. 2 as well.

- Lines 141-142: The statement needs to be corrected, as data shown in Fig. 1 are from P16 animals, and cannot be compared to cell numbers at P6.

- Lines 153-158: The authors' conclusion that cells of the oligodendrocyte lineage at an immature stage are dying is not justified by the data, the TUNEL staining by itself does not reveal which types of cells are labeled. The choice of a single time point and the specific age (P9) is not explained. The relation of these data to the rest of the results is unclear. Does the observation of cell death indicate the induction of death-inducing signaling in NPC1-deficient white matter?

- Lines 167-178, Fig. 3B: The mere presentation of GO analysis results is not sufficient to address the authors' goal "To better understand the pathways associated with oligodendrocyte dysfunction and loss". Here, one would expect a more comprehensive analysis and display of snRNAseq data. This could include volcano plots indicating the most strongly and significantly affected genes in the distinct stages of the lineage together with lineage-dependent, and also -independent (cell-agnostic) pathway analyses. This could also serve to identify candidate pathways to explore the molecular links between NPC1 deficiency and histone hypomethylation (see comment below).

- Lines 192-194, Fig 4: The data shown in Supplemental Fig. 2 should be integrated in Fig. 4 to provide a comprehensive view on the changes. Given previously reported regional differences in white-matter defects of NPC1-deficient mice and patients: would it be possible to provide immunohistochemical staining of different regions to compare the H3K27 modification and myelin damage?

- Lines 179-215: The results are very exciting, and they raise a very important point that should be addressed: the identification of the molecular link between NPC1 deficiency and EZH2-PRC2 dysfunction / histone methylation. In this context, the group's previous observation that defects in oligodendrocyte differentiation and myelination occur also after neuron-specific elimination of *Npc1* suggests that NPC1 deficiency disrupts an intercellular signaling pathway controlling oligodendrocyte differentiation. Authors could select candidate pathways based on previous studies investigating oligodendrocyte development and once again based on a more in-depth *in silico* analyses of their snRNAseq data. Elucidation of pathway(s) linking NPC1-deficiency and epigenetic regulation would represent a very important major advance.

- Lines 216-231: The authors provide first evidence by cell culture experiments that pharmacologic increase of histone methylation at K27 has an impact on NPC1-deficient oligodendrocytes.

Evidently, it would be important for possible translation to clinical trials to show that GSK-J4 mitigates the oligodendrocyte maturation and myelination defects in NPC1-deficient mice *in vivo*. Given its structure, the prodrug should be able to overcome the blood-brain barrier. Once again, the authors should modify the presentation of results. This paragraph somehow stands isolated and interrupts what is presented in the previous (lines 209-215) and next (323-244) "chapter".

- Lines 232-256: The relevance of these data is not clear, and in the present form they are merely confirmatory. A key question is whether treatment of cells with GSK-J4 reverts the observed transcript changes caused by NPC1 deficiency.

- Lines 257-272: The experiments exploring hypomyelination in *Smpd1*-deficient mice are an important element. They support the hypothesis that NPC1 deficiency has a stronger impact on oligodendrocyte development and myelination compared to lack of ASM. The authors should probably mention these experiments in the abstract. Elsewhere, for example in the Discussion, the authors should mention the fact that hypomyelination has been observed in acid sphingomyelinase deficiency patients and in adult *Smpd1*-deficient mice (Buccinna et al., *J Neurochem* 2009; and c.f. Ledesma et al., 2011 *J Neurochem*), the underlying mechanisms and age of onset may of course differ from NPC disease.

- Lines 273-301: The effects of beta-cyclodextrin on NPC1-deficient mice have been documented extensively with various modes of administration. Still, the experiments performed by the authors complement the body of evidence as to this referees' knowledge effects on oligodendrocytes have not been shown previously. Nevertheless, the authors should be careful with their conclusions. Effects of beta-cyclodextrin cannot establish unequivocally a direct role of cholesterol linking NPC1 defects and epigenetic modification. Therefore, corresponding claims should be modified.

Minor points

- Throughout the ms: authors should consider to replace the term "single-nuclear" by "single-nucleus" analogous to "single-cell".

- Throughout the ms: The authors should indicate precisely from which brain region results were obtained. For example, statements such as line 198 "... to isolate OPCs ... from mouse brain..." or line 1044 "Brain sections from ..." must be corrected.

- Lines 86-97: The authors should mention in which brain areas and to which extent precisely hypomyelination has been observed in patients and animals.

- Lines 98-102: To follow up on the previous point, the authors should also explain the rationale/basis why they chose to focus on the forebrain.

- Lines 136, 146, 153: For non-expert readers, references for markers (*olig2*, *sox10*, *ki67*) could be provided.

- Line 226: Authors should consider to integrate supplemental fig 3 in fig 5 as these data are important.

- Lines 397-405: The authors must state the sex of mice used for their study, and they must indicate, whether experiments were performed on littermates following genotyping.

- Line 459 and throughout ms: The term "immunofluorescence" should be replaced by more informative terms (immunocytochemical or -histochemical).

- Line 975: The "absolute" symbol should be around $\log_2(\text{fold-change})$ rather than 0.5. But see comment above suggesting to display numbers of up- and down-regulated genes separately.

- Line 976-978: The description in the legend does not explain correctly what is shown in the figure. Are these false-color micrographs? In any case, display of individual grey-scale channels should be considered, as the blue nuclei are barely visible.

- References: the authors should cite Takikita et al., <https://doi.org/10.1093/jnen/63.6.660>, describing defects in oligodendrocyte development and myelination in the NPC Balb/c model in 2004.

Reviewer #3 (Remarks to the Author):

Cholesterol is known to be an important regulator for CNS myelination, and yet the underlying mechanisms are not fully understood. As reported previously by the investigators here and other groups, mice deficient in *Npc1* displayed hypomyelination during development. The authors further extended their findings here by demonstrating striking transcriptional changes in the oligodendrocyte lineage cells at single cell level as well as increased cell death during OPC maturation process. These changes were associated with loss of repressive histone marks, H3K27me₃, both globally and at promoter regions on neuronal genes.

While the manuscript presented with ample amount of data and detailed characterization of the transcriptomics and epigenetic landscape of oligodendrocyte lineage cells, several major concerns remain in regards to the main conclusion of manuscript.

The use of a global knockout mice greatly diminished the impact of the findings here. *Npc1*^{-/-} mice displayed cellular and molecular deficits in several important cell types in the CNS, including neurons (Yu et al., 2013) and microglia (Columbo et al., 2021). It remains undetermined whether the transcriptional and/or epigenetic alterations shown in the oligodendrocyte lineage cells here were cell autonomous or a consequence of cholesterol mistrafficking from neighboring cells. While no neuronal loss is detected prior to hypomyelination, it may not suggest that it is independent from oligodendrocyte cell loss observed here; as deletion of *Npc1* in neurons only also led to myelin deficits. While it may be technically challenging to quantify cholesterol trafficking across cell types, use of conditional knockout animals specially lacking *Npc1* in the oligodendrocyte lineage cells would greatly help pin down the primary role of *Npc1* in a cell type specific manner. The same notion applies to the rescue of function experiment. While it is convincingly shown that a single i.p. administration of HPβCD partially rescued the hypomyelination phenotype, oligodendrocyte cell loss and transcriptional changes, the specificity of this molecule is undetermined. How can the authors exclude the possibility that the rescue effect is secondary? Two major consequences were observed in *Npc1* deficient mice, enhanced loss of immature and mature oligodendrocytes, and increased expression of neuronal genes in the same population. These seems very confusing. Are these effects independent or synergistic? How come some of the cells underwent cell death while others continued to survive with altered gene expression? This further suggests the effect is not cell autonomous.

One of the major conclusion is *Npc1* is essential for the epigenetic regulation, mainly repressive histone modification H3K27me₃, in oligodendrocyte lineage. However, it remains unclear how NPC1 influences the deposition or the maintenance of this histone mark, despite that the loss of NPC1 led to globally and loci-specific loss of H3K27me₃, and gain of H3K27ac, which may be secondary to the loss of NPC1 in other cell types in the CNS. Furthermore, the analysis of H3K27ac is confusing and perhaps better to be moved to supplementary. What is the rationale for profiling H3K27ac, which is generally considered as an active enhancer mark? Were the levels of H3K27ac increased? Is the expression of the HAT for H3K27ac increased? Do the authors suggest the loss of H3K27me₃ in *Npc1*^{-/-} cells is a consequence of enhanced H3K27ac, or vice versa?

Technical details are missing from the genomic analysis. The numbers of biological replicates (i.e. number of mice, number of samples) in the genomic studies (i.e. snRNA-seq, bulk RNA-seq and ChIP-seq) were not disclosed. It seems these data were collected from one single biological

replicate, which is not acceptable. Second, it is unclear how the correlation of RNA-seq and ChIP-seq data is performed (Fig. 6B and E). What is expressed in y-axis? How is the fold change calculated on H3K27me3 promoter enrichment?

It is unclear how the analysis in Figure 4B is performed. What is relative intensity? How is it calculated? Is it quantified on the entire field or individual cells? It showed H3K27me3 intensity is decreased only in Sox10+ cells, but the images above showed an overall reduction of intensity on all cells in the image.

REVIEWER COMMENTS

Reviewer #1 (Remarks to the Author):

In this study the authors have examined the effects of *Npc1* deficiency on oligodendrocyte lineage cells. Earlier studies from this group have shown that deletion of *Npc1* in oligodendrocytes causes neurodegeneration. In this study, single nuclei RNA-seq (snRNA-seq) of P16 *Npc1*^{-/-} cells from the forebrain revealed a reduction in immature and mature oligodendrocytes but not OPCs. This was verified by immunofluorescence analysis showing that severe reduction in oligodendrocytes occurred between P9 and P16 due to increased cell death. Intriguingly, in addition to reduction in transcripts involved in myelination and cholesterol biosynthesis, genes involved in neurogenesis and synapse formation were upregulated in the mutant immature oligodendrocytes. H3K27me3 histone modification as well as *Ezh2* were reduced in *Npc1*^{-/-} immature oligodendrocytes (and also in OPCs) in vivo and in isolated oligodendrocyte lineage cells. GSK-J4 inhibitor of H3K27 demethylases reversed the proportion of differentiated oligodendrocytes back to wild type levels. ChIP-seq for H3K27me3 and H3K27ac showed reduced H3K27me3 occupancy at the promoters of 16 upregulated genes related to neuronal projection or synapse formation, and of these 10 also had increased H3K27ac occupancy at the promoter. In the final set of experiments, that authors treated *Npc1*^{-/-} mice with a single injection of HPbetaCD at P7 to mobilize stored cholesterol, and they show that oligodendrocytes and myelin were restored, neuronal genes were repressed, and H3K27me3 was increased closer to wt levels.

The authors provide a very interesting set of findings pointing to epigenetic dysregulation in immature oligodendrocytes deficient for *Npc1*, which has important implications in the pathogenesis and pathophysiology of Niemann-Pick disease. They provide a robust set of data to demonstrate defects in repressive histone posttranslational modification in immature oligodendrocytes that are likely to be responsible for the observed upregulation of neuronal genes in mutant oligodendrocytes. The data are clearly presented with adequate statistical analyses, and the manuscript is logically organized. The following are some questions and comments that can be addressed to clarify some ambiguities and strengthen the main conclusions.

1. H3K27me3 is reduced in OPCs and immature O4+ oligodendrocytes in *Npc1*^{-/-} brains. However, there is not much phenotypic change in *Npc1*^{-/-} OPCs, despite detection of *Npc1* transcripts in OPCs in previously published databases at least at a level that is comparable to if not higher than that in immature or mature oligodendrocytes. Do the authors think that the changes in H3K27me3 are already affecting the function of OPCs even if they do not detect obvious changes in their number or proliferation?

This is an excellent question. Of the 48 genes that are differentially expressed in OPCs, 42 (87.5%) are significantly altered in immature oligodendrocytes. This indicates that the deficits

we see in the oligodendrocyte lineage begin in OPCs and continue to worsen as cells differentiate. Additionally, those 48 DE genes in OPCs are enriched for the GO term “neuron projection morphogenesis”, again suggesting the occurrence of functional effects in OPCs. This information has been added to the Results when discussing Fig 3A and is now highlighted in the revised Discussion.

2. If altered cholesterol trafficking is upstream of epigenetic changes, how does one explain the reduced H3K27me3 in OPCs? In Figure 7, mobilizing cholesterol systemically in the mutant mice restores the level of H3K27me3, suggesting that cholesterol mobilization defects are upstream of and lead to reduced H3K27me3. While cholesterol is present in all cells, it is not until the cells reach the immature oligodendrocyte stage that cholesterol biosynthesis becomes prominent and is critically needed for subsequent maturation into myelinating oligodendrocytes. How do the authors think H3K27me3 is downregulated in OPCs (i.e. is it due to altered cholesterol trafficking in OPCs or some other mechanism)?

We agree that this is an important question. As highlighted above, our gene expression analysis of OPCs indicates the occurrence of alterations in the oligodendrocyte lineage that begin early, a finding highlighted in the revised manuscript. We have also added new data (Fig 8) demonstrating that cell autonomous effects of *Npc1* deletion in oligodendrocytes trigger reduced H3K27me3. Additionally, the revised Discussion now highlights recent studies demonstrating that forebrain oligodendrocyte lineage cells rely more on exogenously derived cholesterol than on cholesterol biosynthesis. The bioavailability of cholesterol after endocytosis is disrupted by NPC1 deficiency, and we show that these regional differences in oligodendrocyte lineage cells correlate with the occurrence of H3K27me3 defects in *Npc1*^{-/-} mice (new Figure S5). Finally, we have delved more deeply into the analysis of our expression data from O4+ cells. We now include iPathway analysis (Figure S9) which nominates several signaling pathways that may connect impaired cholesterol trafficking to epigenetics changes. These issues are highlighted in the revised Discussion.

3. Related to question #2, does treating PDGFRalpha-positive purified OPCs (not immature oligodendrocytes) with HPbetaCD or a similarly acting agent also restore H3K27me3 levels?

We were also quite interested in this question and attempted several times to treat OPCs in vitro with varying doses of HPβCD. Unfortunately, we found the cells to be quite sensitive to this treatment and died. Doses low enough to be non-toxic were too low to have an effect, but we do not believe these data are interpretable as these doses were 10-100 fold lower than what is effective in mobilizing stored cholesterol other NPC1 deficient cell types, including primary neurons.

4. Were neuronal genes also upregulated in OPCs, in addition to immature oligodendrocytes?

Yes. As indicated in the answer to question 1, we have highlighted this in our revised Results, in Figure 3A, and in the revised Discussion.

5. I am trying to understand if all the observed epigenetic dysregulation and neuronal gene expression can be simply attributed to trafficking defects of cholesterol, i.e. altered level and location and function of cholesterol in the cells, or whether *Npc1* defects in OPCs can lead to alterations in major signaling pathways that are critical for the terminal differentiation of oligodendrocytes, rather than maturation defects directly due to altered cholesterol trafficking. The authors discuss this a little in the Discussion when they allude to mTOR and TFEB signaling, but they mix up the function in different stages of oligodendrocyte lineage cells, making it difficult to decipher how they are interpreting the OPC data. This question is stemming from the paucity of data on the critical role for cholesterol in OPC-to-oligodendrocyte differentiation, compared to the well documented role for cholesterol in newly formed oligodendrocytes transitioning into mature myelinating oligodendrocytes. The data are not entirely consistent with the exclusive role of *Npc1*^{-/-} defect – cholesterol defect – H3K27me3 transcriptional repression in immature oligodendrocytes. H3K27me3 changes may already be present prior to the stage at which cholesterol is considered critical.

We apologize for any error in our prior Discussion of the role for mTOR and TFEB signaling in oligodendrocyte differentiation. This is an important issue and we have worked diligently to address the Reviewer's comments in the revised manuscript. As indicated in our answer to question 2, our gene expression analysis of OPCs indicates the occurrence of alterations in the oligodendrocyte lineage that begin early, a finding highlighted in the revised manuscript (Fig 3). We have also added new data (Fig 8) demonstrating that cell autonomous effects of *Npc1* deletion in oligodendrocytes trigger reduced H3K27me3. Additionally, the revised Discussion now highlights recent studies demonstrating that forebrain oligodendrocyte lineage cells rely more on exogenously derived cholesterol than on cholesterol biosynthesis. The bioavailability of cholesterol after endocytosis is disrupted by NPC1 deficiency, and we show that these regional differences in oligodendrocyte lineage cells correlate with the occurrence of H3K27me3 defects in *Npc1*^{-/-} mice (new Figure S5). Finally, we have delved more deeply into the analysis of our expression data from O4+ cells. We now include iPathway analysis (Fig S9) which nominates several signaling pathways that may connect impaired cholesterol trafficking to epigenetics changes. These issues are highlighted in the revised Discussion.

6. The neuronal genes whose upregulation in *Npc1*^{-/-} cells is reversed by HP@CD seem to be already expressed, albeit at low levels for some, in wt oligodendrocyte lineage cells. They may belong to a family of neuronal genes that are transcribed in OL lineage genes at low abundance but probably not at a functionally significant level. It is possible that upregulation of these genes occurs in *Npc1*^{-/-} because the chromatin is already somewhat accessible. Would the authors please provide information as to what proportion of the neuronal genes upregulated in *Npc1*^{-/-} immature oligodendrocytes are detected in wt cells and what proportion of the neuronal genes are completely transcriptionally silent/more tightly repressed in wt cells? It would be interesting

to know if the epigenetic dysregulation only affects neuronal genes in oligodendrocytes that are incompletely repressed.

All of the neuron-related genes listed in our dataset are expressed at detectable levels in at least a portion of both WT and *Npc1*^{-/-} cells. It is an intriguing possibility that initial chromatin accessibility contributes to their susceptibility to epigenetic dysregulation, but we cannot definitively determine this from our analysis.

7. Would the authors speculate on whether or how their finding that specific neuronal genes are upregulated in *Npc1*^{-/-} immature oligodendrocytes could potentially be linked to the documented neuronal dysfunction in NP disease?

In new data (Fig 8) we demonstrate using *Npc1* conditional null mice that diminished H3K27me3 in oligodendrocyte lineage cells occurs cell autonomously: it is seen after *Olig2*-Cre but not after *Syn1*-Cre mediated *Npc1* gene deletion. The upregulation of neuron genes in oligodendrocyte lineage cells may have important functional consequences, such as increased susceptibility to excitotoxicity. This is an issue that we are just beginning to explore experimentally and hope to address in a follow up study.

Overall, this is a very interesting manuscript that opens up many additional questions that could be explored experimentally.

Thank you for the positive comments.

Reviewer #2 (Remarks to the Author):

Kunkel and colleagues studied molecular links between oligodendrocyte development, cholesterol and Niemann-Pick Type C (NPC) disease. In this rare autosomal-recessive lysosomal storage disorder, defects in either NPC1 or NPC2 protein impair the exit of unesterified cholesterol from late endosomes/lysosomes and cause accumulation of this and other lipids within the organelle. At present, it is a veritable mystery how this well-described cellular defect provokes distinct forms of the disease with age of onsets ranging from perinatal to adult and symptoms ranging from perinatally lethal liver failure to neurologic/psychiatric decline progressing over years. *Npc1*-deficient mice recapitulate key visceral and neurologic symptoms and die prematurely. As shown previously by many groups including the authors, NPC1 deficiency impairs the development of oligodendrocytes (Yu & Lieberman, 2013; Takikita et al., 2004; De Nuccio et al., 2019), and leads to hypomyelination (e.g. Palmeri et al., 1994; Higashi et al., 1995; Walterfang et al., 2010) in animal models and patients. However, the mechanisms have remained unknown. Despite substantial advances, no curative therapy is currently available or under clinical development. Therefore, new mechanistic insight and new therapeutic targets are

highly welcome.

The work by Kunkel and colleagues has the potential to deliver on this: it suggests a new epigenetic mechanisms that impairs myelination in NPC1-deficient mice and that may serve as new drug target. Their work has the great merit to start by an unbiased examination of brain cells and their reaction to NPC1 deficiency using single-cell/nucleus transcriptome analyses. Surprisingly, only one publication so far reported NPC disease-related results obtained by this fertile approach (Cougoux et al., 2020), which makes the authors' data all the more interesting and valuable. The present work shows that the defect in oligodendrocyte development in NPC1-deficient mice is caused by decreased levels of histone methyltransferase EZH2, which is part of the polycomb repressor complex 2, and the transcriptional repressor H3K27me3. The involvement of H3K27me3 in oligodendrocyte development has been discovered within the last years (see Wang et al., 2020 Cell Rep; Bartosovic 2021 Nat Biotech; Zhang 2022 Nat Commun), but the evidence that this pathway is perturbed in NPC disease is new and unexpected. The authors show in vitro that pharmacologic inhibition of specific demethylases reverts the differentiation defect. And they suggest that the defect is specific to NPC1 deficiency and can be reverted in mice by treatment with beta-cyclodextrin.

The results presented by Kunkel and colleague are potentially of high significance for the Niemann-Pick field, and beyond, notably for diseases where defects in epigenetic regulation of brain cell differentiation may destabilize neuronal circuits and provoke their dysfunction. The methods are sound and described in sufficient detail except for instances indicated below. However, in its present form the manuscript is unsuitable for publication, and requires a major overhaul. Several conclusions are not supported by the data, and key questions need to be addressed experimentally to mark a significant advance. The analysis and presentation of data must be improved, and the organisation of the manuscript needs to be revised. The authors should consider the following points:

Major points

- The authors should revise their Abstract, Introduction and Discussion to explain more succinctly the overall goal of the study. In its present form, the declared aim seems to oscillate between "mechanisms of oligodendrocyte development" and "disease mechanisms in NPC". Notably the Introduction exposes fundamental aspects of oligodendrocyte development, although factually the work aims at disease mechanisms in NPC disease (see first paragraph of Results and Discussion), evidently with a focus on oligodendrocytes. In any case, the authors should also consider the following when revising the Introduction

We thank the Reviewer for the positive comments on our work and the constructive suggestions to improve the presentation of our results. Detailed information on the changes made to the revised manuscript are indicated below.

- - Lines 61-68: The description of oligodendrocyte precursor cells needs to be revised and the reviews updated. The term "oligodendrocyte precursor cells" is often used in an ambiguous

manner, designating precursors of oligodendrocytes, but also resident cells within the brain (NG2-positive), whose function is still debated. In any case, the field dealing with this cell type(s) has seen a revolution within the last decade, which should be acknowledged by the authors by citing more recent articles/reviews.

We completely agree with this comment and have revised the indicated sentences in the Introduction. We now indicate that OPCs are a heterogeneous group of cells that retain some pluripotency and may have varying functions, including antigen presentation. Updated references have been added to support this statement.

- - Lines 69-77: This paragraph should be revised, as it confounds several topics and it does not reflect the current state of knowledge. For example, the first statement concerning cholesterol and oligodendrocytes differentiation is unclear. Can cholesterol be considered a signal controlling oligodendrocytes differentiation? One gets the impression that the authors confound oligodendrocyte differentiation with myelin biogenesis. These are distinct developmental processes.

We thank the Reviewer for these comments, and highlight cholesterol's importance in myelination rather than oligodendrocyte differentiation.

- The use of single cell/single nucleus analyses has become a method of choice to identify cell type-specific reactions to disease, and therefore the authors' data are highly welcome as new source for subsequent analyses in the field. However, the explanation and display of results including quality control data and differently expressed genes should be improved to follow established standards. Moreover, the data are not explored to reveal their full potential. Some suggestions:

- - The authors should show quality control-relevant data in main or supplementary figs. This includes histograms with the frequency distributions of cellular and gene UMI counts as well as mitochondrial gene counts together with the chosen cutoffs. They should further show the violin plots that are mentioned in line 558.

We apologize for the omission of this information from the original submission. This information is now explicitly discussed in the Methods and plots of the data have been added to the supplement as new Fig S10.

- - Lines 536-549: It is unclear whether single nuclei from individual animals were pooled or whether they were bar-coded to track interindividual variation and batch effects. This should be explained.

Once again, we apologize that this was not clearly presented in the original submission. We have added information on replicate numbers to the Results and Methods. Our snRNA-seq and bulk

RNA-seq experiments used 3 WT vs 3 *Npc1* null mice. All samples were barcoded to track variation.

- - Lines 581-582: The authors should explain how marker genes of cell types were chosen. Reference(s) relevant to their phrase "previously determined" should be given.

As requested, references for this have been added.

- - Fig. 1A. The authors should show in Fig. 1 UMAP plots not only for wildtype but also for mutant mice to allow for visual comparison of cell composition. The axes should be labeled according to standards indicating the type of plot. The colors in the legend boxes are barely visible, the authors could increase the size of the boxes.

We have revised Fig 1A as requested. The UMAP has been split based on genotype and the axes properly labeled. The size of the color/legend has been increased. Thanks for the suggestions.

- - Fig. 1B: The authors should consider to replace or complement the bubble matrix by more informative plots displaying expression levels of selected marker genes in the different cell clusters. For example, violin plots reveal the distribution of values in each cluster in a more accessible manner (color coding in bubbles is difficult for some readers).

The bubble matrix was kept in panel 1B because it efficiently relays multidimensional information. A heatmap displaying marker gene expression across cell types has been added as Fig S1.

- - Fig. 1C: The authors should expand the figure and show numbers of both up- (positive) and down-regulated (negative) genes separately.

Fig 1C is revised as suggested.

- - Lines 126-129, Fig. 1D: The authors should represent the percentages of cells from all individual clusters in wildtype and mutant animals using stacked bar/column plots in order to provide a more detailed view on the changes across cells.

The stacked bar column has been added to Fig 1D, with an insert highlighting oligodendrocyte lineage cells.

- Lines 129-167, Fig. 1D,E & Fig. 2, 3: These parts of the ms should be reorganized, as the sequence of results presented and their display in figures is confusing. The complementation/validation of snRNAseq data is an important aspect. On the other hand, as stated by the authors, their data confirm previously published results. Therefore, in these paragraphs and figures the authors should clearly indicate what has been shown in previous

publications and what is new. Evidently, the focus should be on new results. To this end, experiments bringing new insight would be immunohistochemical staining of stage-specific markers to distinguish myelinating from non-myelinating oligodendrocytes. These markers should probably include those used to identify the different stages in their snRNA-derived cell clusters (see Fig. 1B, and PCR targets: lines 159-167; see also below). The choice of ages that were investigated should be explained. To further streamline this part, authors should move panel E from Fig. 1, where it is misplaced, to Fig 2. Further, the PCR experiments (Lines 159-167; Fig. 3A) performed on forebrain lysates stand isolated. The indicated rationale for the experiments "to further analyse the oligodendrocyte lineage" is rather weak and out of context. These results are part of the results presented in the paragraphs before and should be integrated after line 134. Accordingly panel A of fig. 3A should be moved to Fig. 2 as well.

As suggested, old Figures 1E and 3A have been incorporated into new Figure 2 as panels 2A and 2E, respectively. The text has been streamlined and we focus on new results. New panel 2A is retained to provide the reader with a convenient reference, while we acknowledge this is confirmatory for experts in Niemann-Pick Disease (while experts in developmental myelination may be less familiar with this observation). We have clarified in the text that the choice in ages was based upon the timing of developmental myelination, with P6 being associated with large numbers of OPCs, P9 as a time of differentiating and immature cells, and P16 as a time of peak myelination with large numbers of myelin forming cells. Finally, we note that we were unsuccessful in multiple attempts to generate quantitative data from immunohistochemical staining using stage-specific markers, although that we agree such data would be informative.

- Lines 141-142: The statement needs to be corrected, as data shown in Fig. 1 are from P16 animals, and cannot be compared to cell numbers at P6.

We agree, and the text has been modified.

- Lines 153-158: The authors' conclusion that cells of the oligodendrocyte lineage at an immature stage are dying is not justified by the data, the TUNEL staining by itself does not reveal which types of cells are labeled. The choice of a single time point and the specific age (P9) is not explained. The relation of these data to the rest of the results is unclear. Does the observation of cell death indicate the induction of death-inducing signaling in NPC1-deficient white matter?

We have clarified our writing to indicate that TUNEL staining was performed at P6, P9, and P16. We attempted to co-stain TUNEL positive cells with markers of oligodendrocyte lineage cells, but found our staining protocol ineffective when combined with TUNEL staining. Therefore, as suggested, we modified our wording to indicate that our data suggest death of oligodendrocyte lineage cells. We note that these TUNEL positive cells are confined to white matter at P9, suggesting that it is the immature oligodendrocytes that are dying.

- Lines 167-178, Fig. 3B: The mere presentation of GO analysis results is not sufficient to address

the authors' goal "To better understand the pathways associated with oligodendrocyte dysfunction and loss". Here, one would expect a more comprehensive analysis and display of snRNC data. This could include volcano plots indicating the most strongly and significantly affected genes in the distinct stages of the lineage together with lineage-dependent, and also - independent (cell-agnostic) pathway analyses. This could also serve to identify candidate pathways to explore the molecular links between NPC1 deficiency and histone hypomethylation (see comment below).

As suggested, volcano plots have been added as a new Fig 3B and the top DE genes annotated. Moreover, we performed GSEA analysis of the snRNA-seq data for the oligodendrocyte lineage but this did not reveal additional insights, as the top pathways were all the same as the GO terms shown in Fig 3A. We attempted a pathway analysis created for use on single cell datasets, but struggled to find anything of biological relevance. To more completely address the Reviewer's comment, we performed iPathway analysis on bulk RNAseq data derived from O4+ cells. These data are now presented in new Fig S9 and the results incorporated into the revised Discussion.

- Lines 192-194, Fig 4: The data shown in Supplemental Fig. 2 should be integrated in Fig. 4 to provide a comprehensive view on the changes. Given previously reported regional differences in white-matter defects of NPC1-deficient mice and patients: would it be possible to provide immunohistochemical staining of different regions to compare the H3K27 modification and myelin damage?

Quantification from images shown in old Fig S2/new Fig S4 is included in Fig 4B. We found that there is not enough room to include all of the images from the new Fig S4 in the main figure. As suggested, we have added staining for histone methylation in the white matter of the cerebellum in new Fig S5. Interestingly, we did not detect significant changes in histone methylation in this region, which also does not show significant changes in the density of Olig2+ cells (new Fig S3). We thank the Reviewer for encouraging this additional analysis, which suggests that the most severe epigenetic and myelination defects are in the rostral forebrain. This is now addressed in the revised Discussion.

- Lines 179-215: The results are very exciting, and they raise a very important point that should be addressed: the identification of the molecular link between NPC1 deficiency and EZH2-PRC2 dysfunction / histone methylation. In this context, the group's previous observation that defects in oligodendrocyte differentiation and myelination occur also after neuron-specific elimination of *Npc1* suggests that NPC1 deficiency disrupts an intercellular signaling pathway controlling oligodendrocyte differentiation. Authors could select candidate pathways based on previous studies investigating oligodendrocyte development and once again based on a more in-depth in silico analyses of their snRNAseq data. Elucidation of pathway(s) linking NPC1-deficiency and epigenetic regulation would represent a very important major advance.

We thank the Reviewer for the positive comments and suggestion. We now demonstrate using *Npc1* conditional null mice that H3K27me3 defects in oligodendrocyte lineage cells occur in a cell autonomous manner, as they are observed after Olig2-Cre mediated *Npc1* gene deletion but not after Syn1-Cre mediated gene deletion (new Fig 8A, B). Several candidate pathways that might mediate this effect were identified by iPathway analysis of isolated O4+ cells (new Fig S9) and are highlighted in the revised Discussion.

- Lines 216-231: The authors provide first evidence by cell culture experiments that pharmacologic increase of histone methylation at K27 has an impact on NPC1-deficient oligodendrocytes. Evidently, it would be important for possible translation to clinical trials to show that GSK-J4 mitigates the oligodendrocyte maturation and myelination defects in NPC1-deficient mice in vivo. Given its structure, the prodrug should be able to overcome the blood-brain barrier. Once again, the authors should modify the presentation of results. This paragraph somehow stands isolated and interrupts what is presented in the previous (lines 209-215) and next (323-244) "chapter".

We agree that administration of GSK-J4 for therapeutic benefit on developmental myelination is an intriguing possibility. We attempted administration of several doses to neonatal pups, but found that it was not well tolerated and was in fact lethal at higher doses, likely due to widespread effects on gene expression across multiple cell types. We also agree that the flow of data presentation could be improved. We re-ordered the figures, swapping Figs 5 and 6, to address this concern. Fig 5 now shows ChIP-seq analysis that was used to identify differentially methylated and expressed genes, while Fig 6 shows the rescue of gene expression and OPC differentiation by GSK-J4.

- Lines 232-256: The relevance of these data is not clear, and in the present form they are merely confirmatory. A key question is whether treatment of cells with GSK-J4 reverts the observed transcript changes caused by NPC1 deficiency.

We appreciate this comment. To address the Reviewer's question, we treated O4+ cells in culture with GSK-J4 or the inactive GSK-J5. qRT-PCR demonstrates that treatment with GSK-J4 decreases the expression of neuronal genes that are differentially methylated (new Fig 6B).

- Lines 257-272: The experiments exploring hypomyelination in *Smpd1*-deficient mice are an important element. They support the hypothesis that NPC1 deficiency has a stronger impact on oligodendrocyte development and myelination compared to lack of ASM. The authors should probably mention these experiments in the abstract. Elsewhere, for example in the Discussion, the authors should mention the fact that hypomyelination has been observed in acid sphingomyelinase deficiency patients and in adult *Smpd1*-deficient mice (Buccinna et al., J Neurochem 2009; and c.f. Ledesma et al., 2011 J Neurochem), the underlying mechanisms and age of onset may of course differ from NPC disease.

Thank you for the suggestion. Information about myelin defects in ASMD mice and patients has been added to the Results.

- Lines 273-301: The effects of beta-cyclodextrin on NPC1-deficient mice have been documented extensively with various modes of administration. Still, the experiments performed by the authors complement the body of evidence as to this referees' knowledge effects on oligodendrocytes have not been shown previously. Nevertheless, the authors should be careful with their conclusions. Effects of beta-cyclodextrin cannot establish unequivocally a direct role of cholesterol linking NPC1 defects and epigenetic modification. Therefore, corresponding claims should be modified.

We have softened our wording in the Discussion to indicate that our data suggest, rather than establish unequivocally, a direct effect of HP β CD-mediated cholesterol mobilization on epigenetic regulation in oligodendrocytes.

Minor points

- Throughout the ms: authors should consider to replace the term "single-nuclear" by "single-nucleus" analogous to "single-cell".

As suggested, this has been changed throughout the manuscript.

- Throughout the ms: The authors should indicate precisely from which brain region results were obtained. For example, statements such as line 198 "... to isolate OPCs ... from mouse brain..." or line 1044 "Brain sections from ..." must be corrected.

Clarifications of brain regions used for studies have been added throughout the manuscript.

- Lines 86-97: The authors should mention in which brain areas and to which extent precisely hypomyelination has been observed in patients and animals.

We have clarified in our revised manuscript that in NPC patients, myelin loss is most prominent in the major white matter tracts. In *Npc1* deficient mice, hypomyelination occurs across the brain, but is most severe in the forebrain.

- Lines 98-102: To follow up on the previous point, the authors should also explain the rationale/basis why they chose to focus on the forebrain.

As suggested, we have added a statement pointing out that the myelin deficit is most severe in the forebrain and that single-cell sequencing has been previously reported for the cerebellum.

- Lines 136, 146, 153: For non-expert readers, references for markers (olig2, sox10, ki67) could be

provided.

As suggested, we added references for oligodendrocyte marker genes and for KI67 and TUNEL.

- Line 226: Authors should consider to integrate supplemental fig 3 in fig 5 as these data are important.

As suggested, old Fig S3 is now incorporated as new Fig 6A.

- Lines 397-405: The authors must state the sex of mice used for their study, and they must indicate, whether experiments were performed on littermates following genotyping.

As suggested, we added to the Methods: "Littermate mice were genotyped prior to use in experiments. Approximately equal numbers of male and female mice were used in these studies, as a sex-dependent difference in neurological phenotype has not been observed."

- Line 459 and throughout ms: The term "immunofluorescence" should be replaced by more informative terms (immunocytochemical or -histochemical).

All instances of "fluorescence" have been changed, as suggested.

- Line 975: The "absolute" symbol should be around $\log_2(\text{fold-change})$ rather than 0.5. But see comment above suggesting to display numbers of up- and down-regulated genes separately.

Good catch. We have fixed the error.

- Line 976-978: The description in the legend does not explain correctly what is shown in the figure. Are these false-color micrographs? In any case, display of individual grey-scale channels should be considered, as the blue nuclei are barely visible.

These are not false-color micrographs. The DAPI channel has been brightened to make it more visible.

- References: the authors should cite Takikita et al., <https://doi.org/10.1093/jnen/63.6.660>, describing defects in oligodendrocyte development and myelination in the NPC Balb/c model in 2004.

This reference has been added.

Reviewer #3 (Remarks to the Author):

Cholesterol is known to be an important regulator for CNS myelination, and yet the underlying mechanisms are not fully understood. As reported previously by the investigators here and other groups, mice deficient in *Npc1* displayed hypomyelination during development. The authors further extended their findings here by demonstrating striking transcriptional changes in the oligodendrocyte lineage cells at single cell level as well as increased cell death during OPC maturation process. These changes were associated with loss of repressive histone marks, H3K27me3, both globally and at promoter regions on neuronal genes.

While the manuscript presented with ample amount of data and detailed characterization of the transcriptomics and epigenetic landscape of oligodendrocyte lineage cells, several major concerns remain in regards to the main conclusion of manuscript.

The use of a global knockout mice greatly diminished the impact of the findings here. *Npc1*^{-/-} mice displayed cellular and molecular deficits in several important cell types in the CNS, including neurons (Yu et al., 2013) and microglia (Columbo et al., 2021). It remains undetermined whether the transcriptional and/or epigenetic alterations shown in the oligodendrocyte lineage cells here were cell autonomous or a consequence of cholesterol mistrafficking from neighboring cells. While no neuronal loss is detected prior to hypomyelination, it may not suggest that it is independent from oligodendrocyte cell loss observed here; as deletion of *Npc1* in neurons only also led to myelin deficits. While it may be technically challenging to quantify cholesterol trafficking across cell types, use of conditional knockout animals specially lacking *Npc1* in the oligodendrocyte lineage cells would greatly help pin down the primary role of *Npc1* in a cell type specific manner.

The same notion applies to the rescue of function experiment. While it is convincingly shown that a single i.p. administration of HPβCD partially rescued the hypomyelination phenotype, oligodendrocyte cell loss and transcriptional changes, the specificity of this molecule is undetermined. How can the authors exclude the possibility that the rescue effect is secondary?

These are excellent questions that we address with new experimental data in new Fig 8. We now demonstrate using *Npc1* conditional null mice that H3K27me3 defects in oligodendrocyte lineage cells occur in a cell autonomous manner, as they are observed after *Olig2*-Cre mediated *Npc1* gene deletion but not after *Syn1*-Cre mediated gene deletion (new Fig 8A, B). Using *Olig2*-Cre, *Npc1* fl/- mice, we demonstrate that a single administration of HPβCD partially rescues myelination (Fig 8C), *Olig2*⁺ cell density (Fig 8D), and H3K27me3 staining in oligodendrocyte lineage cells.

Two major consequences were observed in *Npc1* deficient mice, enhanced loss of immature and mature oligodendrocytes, and increased expression of neuronal genes in the same population. These seems very confusing. Are these effects independent or synergistic? How come some of the cells underwent cell death while others continued to survive with altered gene expression? This further suggests the effect is not cell autonomous.

Our data indicate that a subset of oligodendrocyte lineage cells die, while remaining viable cells exhibit epigenetic dysregulation and increased expression of neuronal genes. New data in Fig 8 demonstrate that the epigenetic dysregulation occurs through a cell autonomous mechanism. Cell death likely occurs in both a cell-autonomous and non-autonomous fashion, as reduced numbers of Olig2+ cells were observed in both Olig2-Cre and Syn1-Cre mouse brains. We speculate that this cell death may be a consequence of the block in differentiation, as it has been shown that during development, premyelinating oligodendrocytes either mature to form myelin or they degenerate (<https://pubmed.ncbi.nlm.nih.gov/9128255/>). We have incorporated this idea into the revised Discussion.

One of the major conclusion is *Npc1* is essential for the epigenetic regulation, mainly repressive histone modification H3K27me3, in oligodendrocyte lineage. However, it remains unclear how NPC1 influences the deposition or the maintenance of this histone mark, despite that the loss of NPC1 led to globally and loci-specific loss of H3K27me3, and gain of H3K27ac, which may be secondary to the loss of NPC1 in other cell types in the CNS.

As discussed in response to the prior question, our new data in Fig 8 demonstrate that loss of H3K27me3 in oligodendrocyte lineage cells occurs in a cell autonomous manner. To explore possible pathways that link lysosomal defects to this epigenetic change, we now include in new Fig S9 iPathway analysis performed on bulk RNA-seq from O4+ cells. This analysis identified several pathways that may contribute to the pathological defects; this information is now included in the revised Discussion.

Furthermore, the analysis of H3K27ac is confusing and perhaps better to be moved to supplementary. What is the rationale for profiling H3K27ac, which is generally considered as an active enhancer mark? Were the levels of H3K27ac increased? Is the expression of the HAT for H3K27ac increased? Do the authors suggest the loss of H3K27me3 in *Npc1*^{-/-} cells is a consequence of enhanced H3K27ac, or vice versa?

We apologize that our initial presentation of these data was unclear. H3K27ac is included as it often inversely correlates with H3K27me3. Indeed, we searched explicitly for neuronal genes that were over-expressed in mutant O4+ cells and which showed both diminished H3K27me3 and increased H3K27ac in their promoters. We highlight three such genes in the description of our ChIP-seq data (Fig 5F), and demonstrate that the expression of these genes is rescued in O4+ cells in culture following treatment with GSK-J4 (new Fig 6B) and in *Npc1* null mice following treatment with HPβCD (Fig 7F). Therefore, it is our strong preference to keep the H3K27ac data in Fig 5.

Technical details are missing from the genomic analysis. The numbers of biological replicates (i.e. number of mice, number of samples) in the genomic studies (i.e. snRNA-seq, bulk RNA-seq and ChIP-seq) were not disclosed. It seems these data were collected from one single biological

replicate, which is not acceptable. Second, it is unclear how the correlation of RNA-seq and ChIP-seq data is performed (Fig. 6B and E). What is expressed in y-axis? How is the fold change calculated on H3K27me3 promoter enrichment?

It is unclear how the analysis in Figure 4B is performed. What is relative intensity? How is it calculated? Is it quantified on the entire field or individual cells? It showed H3K27me3 intensity is decreased only in Sox10+ cells, but the images above showed an overall reduction of intensity on all cells in the image.

Additional technical and experimental details for the sequencing have been added to the manuscript. snRNA-seq and bulk RNA-seq were performed on 3 mice per genotype. ChIP-seq was performed on cells derived from 6 mice per genotype. All samples were barcoded to track individual variation. Fold change for H3K27me3 was measured by ActiveMotif using their proprietary analytic pipelines.

For Fig 4B, intensity of H3K27me3 staining was measured and normalized to WT, so “relative intensity” is relative to WT levels. Images were captured from 3 locations in the corpus callosum (or cortex for neurons) for each mouse. Methylation intensity was measured only in cells positive for SOX10 or NeuN. The H3K27me3 intensity per cell was averaged for each biological replicate.

REVIEWERS' COMMENTS

Reviewer #1 (Remarks to the Author):

The authors have made an earnest attempt to address all the points I had raised on the previous submission. They could not address all the points, but they provide an additional compelling figure showing the cell autonomous nature of Npc1 defect in altering cholesterol trafficking and subsequent H3K27me3 function. The manuscript is significantly stronger. It is a very exciting and important study that will be of interest to a wide readership.

Reviewer #2 (Remarks to the Author):

The authors have well responded to my comments, and the revision has greatly improved the quality of the ms, which I regard now as acceptable for publication.

Reviewer #3 (Remarks to the Author):

The revised manuscript has been greatly improved with the additional experiments. The inclusion of conditional knockout mice using two separate cell type-specific cre drivers has greatly improve the impact of the findings. I only have minor comments, which the authors need to clarify:

1. Please use standard nomenclature for transgenic mice. It is unclear what "Olig2-Cre; Npc1flox/-" is referring to. If this is the mice expressing homozygous flox/flox alleles, resulting in deletion of Npc1 in Olig2-expressing cells, it should be referred as "Olig2-Cre; Npc1flox/flox"
2. FigS9 should be better explained. It is unclear what X- or Y-axis is referring to.

REVIEWERS' COMMENTS

Reviewer #1 (Remarks to the Author):

The authors have made an earnest attempt to address all the points I had raised on the previous submission. They could not address all the points, but they provide an additional compelling figure showing the cell autonomous nature of *Npc1* defect in altering cholesterol trafficking and subsequent H3K27me3 function. The manuscript is significantly stronger. It is a very exciting and important study that will be of interest to a wide readership.

We thank Reviewer #1 for their comments.

Reviewer #2 (Remarks to the Author):

The authors have well responded to my comments, and the revision has greatly improved the quality of the ms, which I regard now as acceptable for publication.

We thank Reviewer #2 for their comments.

Reviewer #3 (Remarks to the Author):

The revised manuscript has been greatly improved with the additional experiments. The inclusion of conditional knockout mice using two separate cell type-specific cre drivers has greatly improve the impact of the findings. I only have minor comments, which the authors need to clarify:

1. Please use standard nomenclature for transgenic mice. It is unclear what “Olig2-Cre; *Npc1*flox/-” is referring to. If this is the mice expressing homozygous flox/flox alleles, resulting in deletion of *Npc1* in Olig2-expressing cells, it should be referred as “Olig2-Cre; *Npc1*flox/flox”

We appreciate Reviewer #3's comment. The nomenclature used is accurate to the genotype of the mice (each mouse has only one floxed *Npc1* allele and the other allele is either WT or null). The breeding scheme to produce these mice is explained on page 15 of the manuscript.

2. FigS9 should be better explained. It is unclear what X- or Y-axis is referring to.

We thank Reviewer #3 for their comment and agree that Supplementary Figure 9 should be better explained. More details have been added in the figure legend.